



# Flow Structure and Mixing Near a Small River Plume Front: Winyah Bay, SC, USA

Christopher Papageorgiou[1], George Voulgaris[1], Alexander Yankovsky[1] and Diane B. Fribance[2]

[1] School of the Earth, Ocean and Environment, University of South Carolina, Columbia SC, USA

[2] Department of Marine Science, Coastal Carolina University, Conway SC, USA

*Correspondence to*: George Voulgaris (gvoulgaris@geol.sc.edu)

**Abstract.** This study presents a comprehensive analysis of Eulerian data collected in the vicinity of the front of a recently discharged river plume from Winyah Bay, SC, USA. The data presented capture the plume structure and evolution and include high-resolution velocity and temperature time series, supplemented by T-S profiles from a MicroCTD profiler. The

observations identified a pre-existing plume extending to 4 meters, with a water density of 1,023.6 kg m$^{-3}$, laying above denser ambient waters. Upon arrival, the newly discharged plume introduced a fresher layer (1,020.7 kg m$^{-3}$) extending to 2.6 meters, gradually thinning due to radial spreading. The plume's frontal propagation measured at 0.36 m s$^{-1}$ with a calculated Froude number of 1.32, indicating gravity current dynamics. Mixing processes were examined using the available overturn potential energy (AOPE) in the water column as described in Smith (2020). The analysis showed that near the bed, bottom boundary

layer turbulence is the main mixing mechanism both before and after the passage of the front. In the surface layer, before the arrival of the front, mixing is driven by wind-induced shear and overturning. Despite high turbulent kinetic energy dissipation in certain regions, shear-induced mixing within the gravity current was minimal. These findings were reflected in the density diffusivity estimates near the surface that varied from 10$^{-6}$ prior to the arrival of the front, increasing to 10$^{-5}$ very near the front and diminishing to 10$^{-10}$ within the plume despite the high velocity shear observed there. Evidence of internal waves was

observed, particularly in the pre-existing plume, providing further insights into the complex hydrodynamic interactions within river plumes and their role in coastal mixing.

## 1. Introduction

Freshwater plumes are common features in coastal environments, where rivers discharge nutrient-rich freshwater into marine waters. Initially, a plume is discharged during ebb tide and behaves like a jet outflow as it enters the coastal ocean. In the absence

of high winds and strong coastal currents it transitions into a buoyancy-driven surface current that expands under the action of gravity (i.e., gravity wave). Further offshore, and assuming there are no constraints on the flow, the plume will turn until in achieves geostrophic balance and propagates as a Kelvin wave (Münchow and Garvine 1993a). Under realistic conditions, the geometry, structure, and dispersion paths of freshwater plumes depend on wind forcing, river discharge, coastal circulation (including tides), and bathymetry (Horner-Devine et al., 2015). Moderate upwelling-favorable winds drive the plume offshore,

causing it to thin and stretch into filament-like shapes that can extend tens of kilometers (Li et al., 2003; Yankovsky et al., 2022). Conversely, under downwelling conditions, the plume is compressed against the coast, deepening as it is driven parallel to the shoreline (Fong and Geyer, 2002).



Lateral spreading and entrainment play pivotal roles in defining plume dynamics. Spreading stretches the plume horizontally,
leading to thinning and increased velocity, which intensifies shear at the base of the plume. This shear drives mixing, which
incorporates denser ambient water into the plume through entrainment. Entrainment increases the density of the plume, causing
it to deepen and reduce its buoyancy, which may subsequently decrease its lateral spreading rate and overall velocity as it
transitions to subcritical flow (Hetland, 2010; MacDonald et al., 2007). Mixing is a common plume feature occurring mainly at
the plume front and base. Usually, the base of the plume experiences stratified-shear instabilities at the plume / ambient water
interface, which may cause mixing between the two water bodies (Kilcher et al., 2012, Mc Donald et al., 2007). The frontal
region is the most energetic as it forms a convergence zone that extends several meters into the water column. The mixing is
intense (O'Donnell et al., 2008) and can reach dissipation rates on the order of $10^{-4}$ to $10^{-3}$ m² s⁻³ diminishing exponentially to
$10^{-7}$ – $10^{-6}$ behind the front (Orton and Jay, 2005; O'Donnell et al., 2008; Horner-Devine et al., 2013; Delatolas et al., 2023).

Recent advancements in dissipation measurement techniques, particularly near the ocean surface, have greatly improved the
ability to quantify turbulent dynamics. Methods leveraging high-resolution velocity data from acoustic Doppler current profilers
(ADCPs) mounted on surface-following platforms now enable precise estimation of dissipation profiles ($\epsilon(z)$) within the upper
few meters of the water column (Zeiden et al., 2023). Additionally, approaches based on Thorpe sorting (Smith, 2020) offer
robust frameworks for characterizing mixing processes, by linking overturn scales to turbulence dissipation, which is particularly
effective in stratified shear flows. Together, these methodologies provide critical tools for resolving dissipation in dynamic
environments, offering improved understanding of turbulence structure.

This study employs advanced methodologies to investigate near-surface turbulence and mixing dynamics within a freshly
discharged plume as it interacts with an existing water mass in the near-field under light upwelling-favorable conditions. High-
resolution time series of flow, temperature, and turbulence structure, extending close to the sea surface, are analyzed alongside
individual temperature-salinity (T-S) profiles. Dissipation rates ($\epsilon$) are estimated using a combination of acoustic Doppler current
profiler (ADCP)-based techniques and Thorpe sorting approaches. These methods allow for precise characterization of turbulent
signals, mixing processes, and hydrographic characteristics (temperature, salinity, and density) across the plume front. By
integrating these state-of-the-art approaches, this study aims to illuminate the plume's kinematics and assess mixing efficiencies
with unprecedented detail.

## 2. Experimental Setup

### 2.1 Data Collection

Data collection took place outside the mouth of Winyah Bay (WB), an estuary located in South Carolina, USA. It receives water
from a drainage area of 47,060 km², primarily through the Pee Dee (~55%) and Little Pee Dee (~20%) rivers and experiences
predominantly semi-diurnal tides (Kim and Voulgaris, 2005). The data presented here were collected as part of a larger synoptic
survey of the Winyah Bay plume that occurred on board the R/V *Savannah* during the period April 12 to April 15, 2023. This
includes time-series of velocity and temperature profiles, accompanied by distinct CTD and turbulence structure profiles
collected at a site located some 10 km off the mouth of the estuary (Fig. 1) with a mean water depth of 11.5 m. The site was
selected to potentially capture the propagation of a newly discharged plume as it entered the coastal ocean.




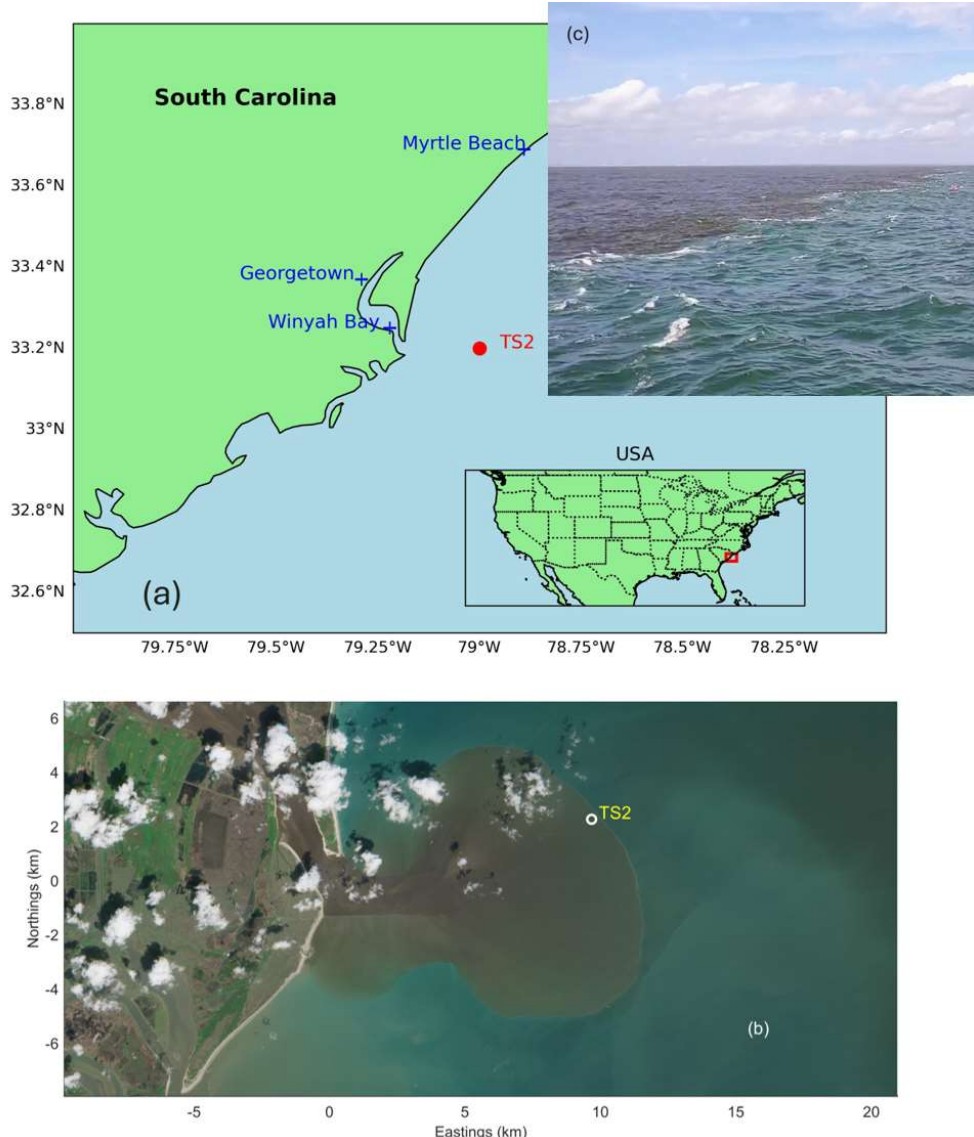

**Figure 1: (a) Location of time series data collection (station TS2) off the coast of South Carolina, USA. (b) Sentinel L2 satellite image (modified Copernicus Sentinel data 2023 processed by Sentinel Hub) of a plume exiting Winyah Bay. Image was captured on 04/15/2023 at 15:49:31 UTC. The TS2 station location is also shown on the image. (c) Photograph of the front examined in this study.**

Two floating platforms were deployed on April 14th, 2023, for a period of ~4 hours capturing data for ~2 hours before and after the passage of a plume front. One platform (RoboCat) was a small catamaran structure equipped with a downward looking 5 beam Nortek Signature 1000 Acoustic Doppler Current Profiler (AD2CP) capable of providing high frequency, high resolution time series of horizontal and vertical currents extending close to the sea surface. The four slanted beams (Broadband, BB mode)



were configured to collect data at 1 Hz over a burst (ensemble) of 8.53 min (i.e., 512 profiles per burst). Cell bin size and range were 0.25 m and 15 m, respectively, and the blanking distance was set to 0.2 m. The 5th beam (HR mode) was sampling simultaneously, recording along-beam, radial velocities at a frequency of 8 Hz resulting in 4,096 radial velocity profiles per ensemble; the HR mode was configured with blanking and bin cell sizes of 0.20 m and 0.04 m, respectively (i.e., 171 cells over
a range of 7 m). The burst data collection was repeated every 9 min resulting in a gap of 0.47 min between successive ensembles. For both BB and HR modes, after accounting for AD2CP transducer submergence, the first bin was located at 0.55 m below the sea surface.

The second platform was equipped with a thermistor array (t-array) consisting of 10 fast responding temperature sensors (RBR
Coda3), arranged on a vertical aluminum rod. The rod was cantilevered from the platform and the individual sensors were separated by 0.3 m. The array provided time-series of temperature in the range extending from 0.02 m to 2.70 m below the sea surface with a sampling frequency of 2 Hz. The data were recorded using a specially built raspberry Pi microcontroller, equipped with a 12 port USB hub providing connections for the RBR units and a GPS unit (for time synchronization).

In addition to the platform, discrete CTD profiles and turbulence dissipation rates were acquired using an uprising MicroCTD profiler (Rockland Scientific) that was manually deployed from a small inflatable boat in the vicinity of the two surface platforms described above. In-situ meteorological data (i.e., wind speed and direction, air temperature, barometric pressure, and relative humidity) were obtained from the meteorological station aboard the R/V *Savannah*, recording at a rate of 1 sample every 30 seconds. Surface wind stress is estimated using the Coupled Ocean-Atmosphere Response Experiment (COARE) algorithm
version 3.6 (COARE 3.6, Fairall et al. 2003, Edson et al. 2013). The inputs to the algorithm included the recorded wind speed and direction, after they were averaged over periods of 512 s to match the AD2CP ensemble averaging scheme, and ship collected information of air and sea temperature, relative humidity, and barometric pressure. A downward IR flux of 400W/m$^2$ and a maximum day time solar radiation of 900 W/m$^2$ were assumed. It should be noted that during data collection the vessel was stationed farther away from the AD2CP and thermistor array, but in proximity to the study site, so as not to interfere with the
flow in the deployment site.

Although no direct observations of the spatial extent of the plume exist, a Sentinel image (see Fig. 1b) obtained on April 15, 2023, at 15:49:31 UTC, ~2 hours after ebb provides a glimpse of the plume geometry. The imagery time corresponds to a similar tidal stage as that experienced during data collection and under similar wind forcing. The freshwater plume and associated front
are clearly shown to be radially dispersing and at the time of the imagery its dimensions were ~ 11 x 7 km. The area of the plume ($A_p$) was estimated to be approximately 62 km$^2$.

## 2.2 Turbulence Measurements

High resolution T-S and velocity shear profile data were collected using a Rockland Scientific microstructure profiler (MicroCTD) equipped with two piezoceramic shear probes, augmented by a fast thermistor, microconductivity sensor, and a
JAC CT sensor for CTD measurements. The profiler was used in an uprising mode providing raw data from approximately 9 m all the way to the sea surface. The CTD sampling frequency was 64 Hz corresponding to a vertical resolution of ~0.01m. A total of 16 individual profiles were collected and the temperature and conductivity measurements were synchronized using the actual uprising velocity and following the procedure recommended by the manufacturer. Discrete dissipation estimates from the



MicroCTD deployments were obtained from the two perpendicular shear probes, using the Rockland Scientific provided processing tools (Lueck, 2016) with default cutoffs for spectral integration using the Nasmyth spectra vs. fitting to the inertial subrange at dissipation rates greater than $1.5 \times 10^{-5}$ W kg$^{-1}$. Each spectrum was the average of individual spectra obtained from 1 s segments with 50% overlap. To maximize vertical resolution, each estimate represents a 2 s record (approximately 1.2 m), with 50% overlap between estimates. The FFT length for ensemble averaging within a spectral estimate was 1 s. The minimum depth for evaluation and minimum vertical velocity were 0 m and 0.5 m/s. Terminal speeds were approximately 0.6 m/s.

Dissipation estimates corresponding to periods of high vibration were manually removed and the spectra from each shear probe were averaged prior to estimating a dissipation value.

In addition to the discrete dissipation profiles obtained with the MicroCTD, more regular dissipation profiles were estimated from the AD2CP HR radial velocity records using the structure function (SF) method (Wiles et al., 2006; Zippel et al., 2018; Scannel et al., 2017) following the process described in Zeiden et al. (2023). This method is suitable for use with moving

platforms as it is immune to sensor motion (Thomson et al., 2019). According to this method, the second-order structure function ($D$) is defined as

$$D(z,r) = < \left( u'_r(z - r/2) - u'_r(z + r/2) \right)^2 > \qquad (1)$$

where $r$ is the spatial distance between measurements, $< >$ denotes time averaging, and $u_r'(z)$ is the demeaned, instantaneous along-beam (radial) velocity. $D$ is then related to the dissipation rate ($\varepsilon$) through the equation:

$$D(z,r) = C_v^2 \varepsilon(z)^{\frac{2}{3}} r^{\frac{2}{3}} + N(z) \qquad (2)$$

where $N$ is an offset representing the noise level in the measurements and $Cv^2$ (= 2.2) is a constant (Wiles et al. 2006). A least-squares linear regression of $D(z, r)$ against $r^{2/3}$ at each elevation z below the sea surface is used to provide estimations of $\varepsilon$ and $N$, with the latter being an indicator of the accuracy of the estimate.

The data were pre-processed following Zeiden et al. (2023) using their MATLAB® code. The pre-processing included removal

of bad velocity values defined as those with low amplitude (< 40 dB) and/or low coherence (< 50). In addition, spikes in the data were removed using an along-beam median filter. Finally, wave bias was removed using an Empirical Orthogonal Function (EOF) based filter that removes profiles of wave orbital velocities estimated from the 4 larger EOFs. Following Zeiden et al (2023), in our analysis we used separation scales ($r$) ranging from one to four times the cell size (i.e., 0.04 m to 0.16 m). Additionally, in the post-processing flow, the resulting dissipation profiles were filtered based on a mean-square percent error

(MSPE) of the least square fit of equation 2; only estimates with MSPE <5% were accepted. More details on the processing method can be found in Zeiden et al (2023).

## 3. Results

### 3.1 Environmental Conditions




The Pee Dee average discharge during data collection was ~550 m³/s (USGS Station 02135200), however given that the Pee Dee

River contributes ~55 to 65% of the fresh water discharged by Winyah Bay (Kim and Voulgaris, 2005) the actual discharge from

the mouth of the estuary was likely at least 1.5 times larger (i.e., ~825 m³/s). Tidal surface elevation data (not shown here) were

available from NOAA station 8661070 (Springmaid Pier, SC) located ~56 km northeastward from the WB mouth. Data collection

started at low water (LW) and proceeded up to ~ 2 hours before high water spanning the period LW to LW+4. During data

collection wind direction was persistent toward the NE (upwelling favorable) with a mean wind speed and surface stress of 5.4

m/s and 0.031N/m², respectively (Fig. 2). The maximum wind stress observed during the sampling period was at ~15:00 UTC,

0.053 N/m². Wave conditions were mild with wave heights ranging between 0.3 – 0.45 m and of very short-wave lengths (see

Fig. 1).

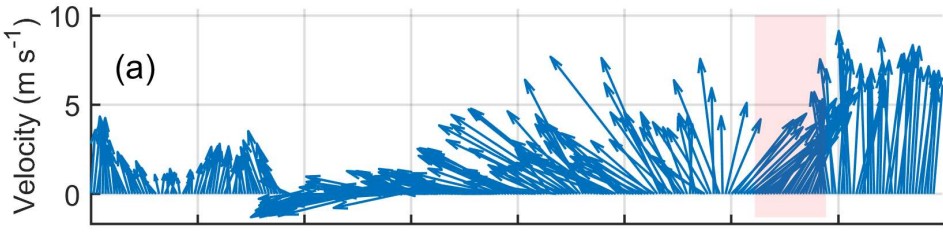

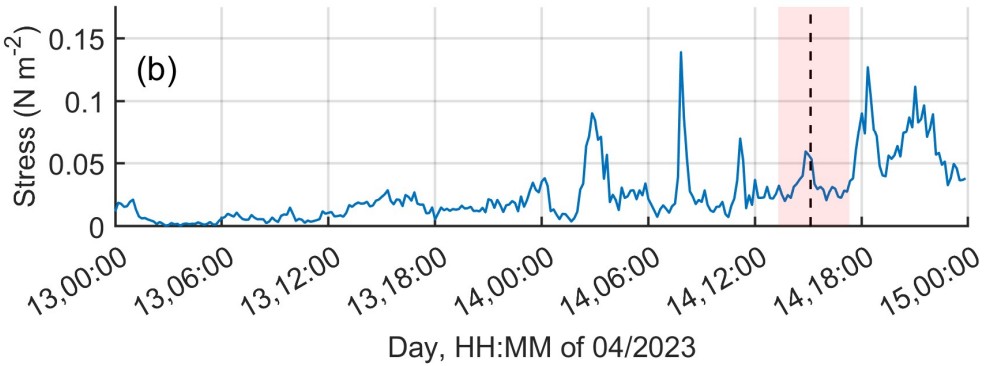

**Figure 2: Time series of: (a) wind vectors for 2 days around the data collection period (red shaded area); and (b) total wind stress estimated using the COARE 3.6 algorithm (black dashed line represents the time of plume front arrival).**

**3.2 Near Surface Temperature Structure**

The time series of both raw (2Hz) and smoothed (using a 180-point (90 s) moving averaging window) thermistor array data are

shown in Figs. 3a and b, respectively. The same smoothed data, gridded as a function of time and elevation, are displayed in the

form of a contour map in Fig. 3c. Initially the near surface temperature ranged from 18.5 °C at the surface to 17.5 °C towards the

bottom (2.82 m below the sea surface). The arrival of the plume front (15:07 UTC time) is identified as the time when all

temperature time series from all depths collapse and shift to a higher temperature. This time corresponds ~1.9 hours after LW.



The mean temperature over the top 3 m of the water column shifted from 18.3 to 18.7 °C, indicating the newly arrived plume

being 0.4 °C warmer than the pre-existing water mass. The near surface mean temperature gradient, defined as the difference

between the bottom and top sensor temperatures, divided over their separation distance (2.82 m), was ~0.4 °C /m until some 15

min prior to the arrival of the front. Then it dropped to 0.1°C/m, increasing over time to almost 1.0 °C/m toward the end of the

data collection period, some 2.4 hours after the plume front passage (see Fig. 3c). As seen from the spacing of the 4 upper

thermistor time series (Fig. 3b), the local vertical temperature gradient within the top 1 m of the water column increases from

0.1 °C/m near the front to 1.4°C/m some 2 hours after the plume.

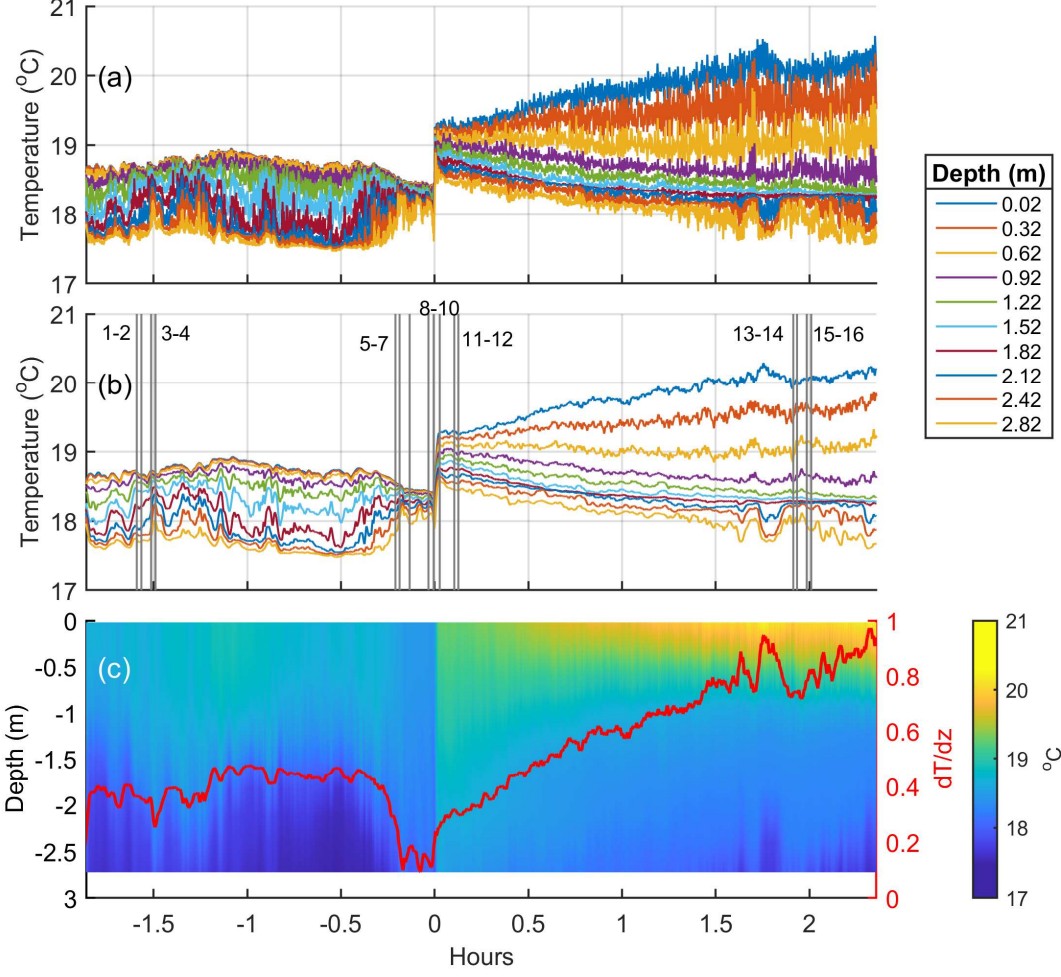

Figure 3: (a) Time series of the raw 2Hz recorded temperatures for each individual thermistor located at depths varying from
0.02 to 2.72m (see legend). (b) Same data as (a) but smoothed using a 180 point (90 s) moving average window. (c) Smoothed
thermistor data (as in (b)) shown as a colored contour plot with temperature gradient (dT/dz) superimposed (red line). The
vertical lines in (b) identify the times of MicroCTD data collection with the numbers adjacent to them denoting the cast number.
Times are relative to the time of plume front arrival.



It is worth noting the oscillations shown in the temperature record in water depths 1.82 to 2.42m during the periods -1.6 to -1.0 hours. They have an amplitude of ~0.2 °C and are visible in both the raw and smoothed data. Similar oscillations are seen later in the record after the plume front has passed though (see times 1.5 to 2 hours). These latter oscillations seem to be limited to

the lowest three sensors corresponding to depths > 2m. These are evidence of potential internal wave activity, something observed in the acoustic backscatter records (see section 4.4) but also identified in other field expeditions in the area.

### 3.3 Temperature – Salinity - Density Profiles

A total of 16 MicroCTD profiles were collected during the experiment and the temperature and salinity data collected were used to estimate water density using the TEOS-10 (Thermodynamic Equation of Seawater, 2010) model. Some profiles were obtained

very close in time, making the temporal distribution of the casts highly irregular, as shown in Fig. 3b. Despite their irregularity, separate groups of profiles represent different periods of the plume propagation, as explored further below.

Based on cast time, in relation to the time the front passed over the sensors, the profiles were sorted into 5 distinct groups. Casts 1-4 (group A) represent conditions before (~1.6 hours) the arrival of the front, when the plume's leading edge was at some distance from the data collection site, while casts 5 – 7 (group B) represent conditions just before (~10 min) the plume's front

passed over the sensors. This is followed by casts 8 –10 (group C), captured when the plume front was exactly over the station. After the front's passage, casts 11-12 (group D), provide an insight of the conditions in the interior of the plume but near the front, some 5-6 minutes after. Finally, casts 13 – 16 (group E), provide the structure of water properties from the interior of the newly arrived plume well after (~2 hours) the front had gone by.

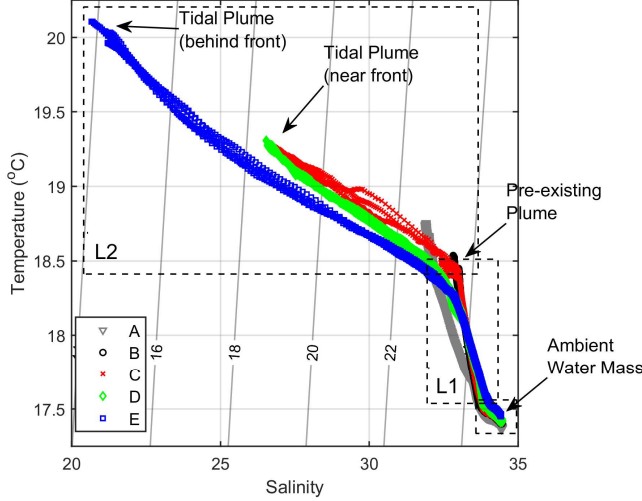

**Figure 4: T-S diagram of the MicroCTD profile data collected during this study color-coded according to the group (A to E) they belong to (see text for details). The three dashed line boxes outline regions of the T-S space corresponding to the newly arrived layer (L2, top left), pre-existing (L1, middle box), and ambient (bottom right) water masses. End-members are indicated with arrows.**





The raw MicroCTD data are shown in a T-S diagram (Fig. 4) with different colors for different groups. Three distinct water
masses can be identified corresponding to: (i) the ambient deep water, (ii) a slightly lighter water mass corresponding to a pre-
existing water mass, presumably from the previous tidal cycle; and (iii) that of the newly arrived, warmer fresher plume. For the
most part, each water mass is characterized by a certain range of T and S with these two parameters exhibiting an almost linear
T-S relationship with different slopes for each water mass.

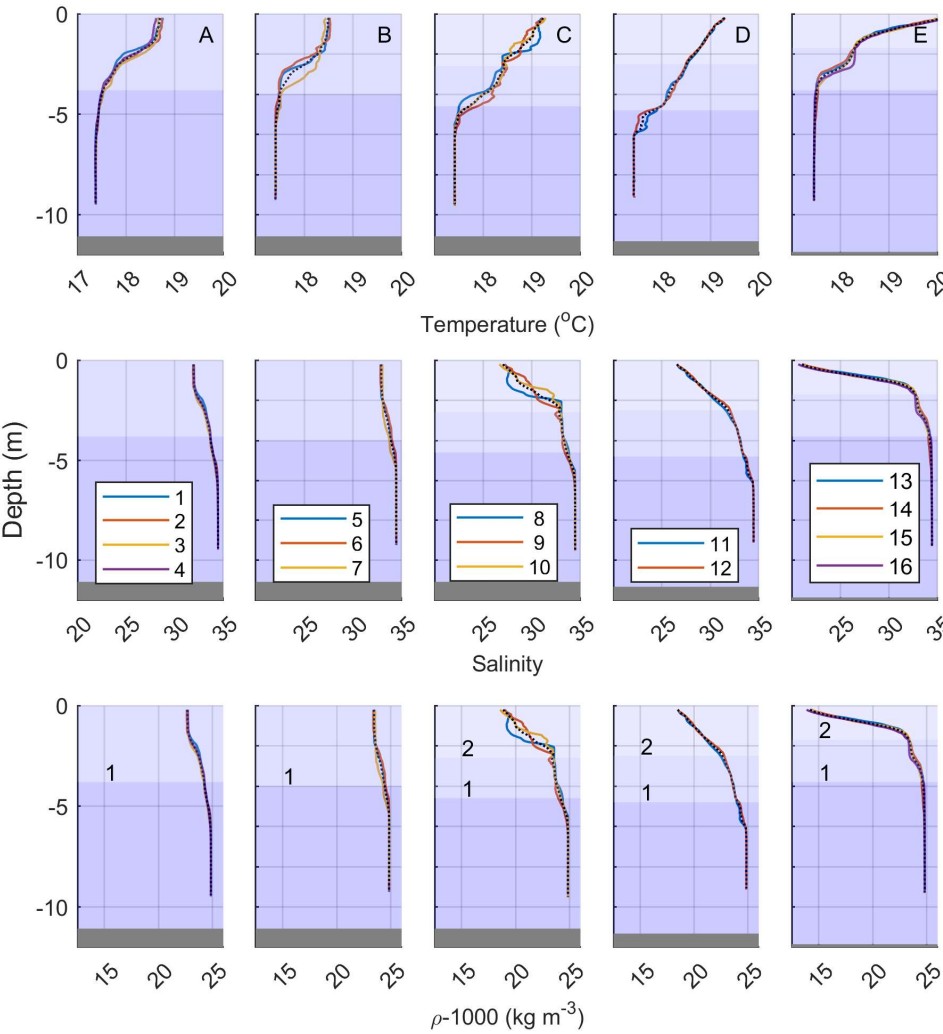

Figure 5: Individual profiles (1 to 16) of temperature (T), salinity (S), and density (ρ) arranged in different panels by group (i.e.,
5 columns for groups A to E). Profiles within each group correspond to different location / distance from the plume front (see
Fig. 3b). Black dashed lines indicate averaged profiles for each group. Different shades of blue, from darker to lighter represent
the different water masses: ambient, pre-existing plume (layer 1), and new plume (layer 2), respectively. For the depths of each
layer see discussion in section 4.





The vertical extent and structure of these water masses are shown in the individual profiles of temperature, salinity, and derived

190 density shown in Fig. 5 for each group separately. Group A and B profiles exhibit a consistent single step structure, while group C profiles show a two-step pattern that appears to persist in the profiles corresponding to groups D and E. These patterns are present in both temperature and salinity, something not surprising given the linear relationship between T and S (see Fig 4). Initially, in Group A and B the uppermost layer (<2 m) displayed temperatures, salinity, and density levels of approximately 18.7 °C, 32, and 1,023 kg m$^{-3}$, respectively. Meanwhile, the ambient bottom layer conditions (>6 m) remained relatively stable, with

195 temperatures ~17.3 °C, salinity 34.4, and density 1,024.8 kg m$^{-3}$. This is also clearly delineated in the T-S diagram shown in Fig, 4. As the front passed through, a sudden rise in surface temperature is observed by approximately 0.7 °C; the water temperature kept increasing with time until the end of data collection (a total of 1.5 °C increase over a period of ~ 2 hours). A similar abrupt shift toward lower values was observed in salinity. At the time of the front's arrival (Group C) salinity was reduced by approximately 5, eventually reaching a decrease of 12. In terms of density, the arrival of the front was associated with a decrease

200 of 4 kg m$^{-3}$, reaching densities of 1,019 kg m$^{-3}$near the sea surface (<2 m). In the interior of the plume (Groups D and E) fresher water was recorded with surface water densities close to 1,014 kg m$^{-3}$.

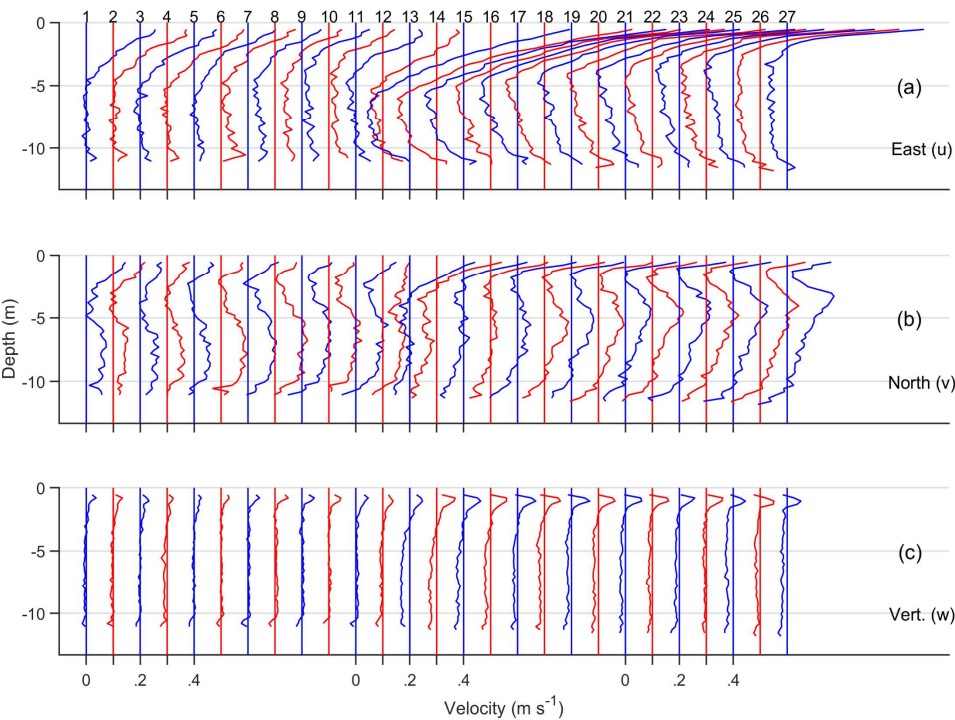

**Figure 6: Individual time averaged current profiles of (a) east (u), (b) north (v), and (c) vertical (w) velocity components. Each profile represents an 8.53-minute average. Vertical lines represent the origin (0 m s$^{-1}$) for each profile; they are offset by 0.10 m s$^{-1}$ and the number next to it represents the profile number.**



### 3.4 Flow Structure

During the ~4 hours of data collection, a total of 27 averaged velocity profiles (ensembles) were collected using the AD2CP. The vertical structure of the horizontal (east and north) and vertical velocity components is presented in Fig. 6, in the form of individual profiles. Profiles 1 to 12 represent flow conditions before the arrival of the front ($t < t_f$), while profiles 14 to 27 represent conditions after that time ($t > t_f$). Profile 13 corresponds to the time when the front ($t_f$) arrived and provides a snapshot of averaged flow conditions within the front.

Overall, prior to the arrival of the front, the velocity of the surface layer (~2 m) is 0.2 m s$^{-1}$ with an ENE direction. Further below, the surface current decays to ~0.1 m s$^{-1}$ and veers towards NNW. After the front passage, both horizontal velocity components increase in magnitude near the sea surface, influenced by the newly arrived plume. Specifically, the near surface east (u) velocity component increases from ~ 0.20 to 0.67 m s$^{-1}$ (see profile 14) and reduces to 0 m s$^{-1}$ at ~4 m depth. This increase in near surface east velocity after the arrival of the plume persists for the remainder of the data collection period, although some 2 hours later the velocity reduces to 0 m/s at much shallower water depths (~ 2m) than those identified prior to the arrival of the plume. The latter indicates a shallowing of the plume, possibly due to radial expansion, but it can also be related to the formation of the trailing front, as described in a semi-analytical model of a radial supercritical plume forced by a constant discharge (Garvine, 1984). The observed flow structure is also consistent with the schematic proposed by Luketina and Imberger (1987, their Figure 2). The vertical component of the flow exhibits consistently positive (upward) flows close to the sea surface and after the plume's arrival negative velocities below the plume that persist all the way to the seabed. The origin of the positive vertical flow at the head of the advancing front is not obvious and contradicts previous observations (O'Donnell et al., 1998). This flow seems to correlate with the strong near-surface horizontal flow within the plume and it turns to negative (downward) at the same elevation the horizontal velocity changes direction below the plume. It should be noted that the same pattern was observed on the radial velocities of the 5$^{th}$ beam (see section 3.6).

### 3.5 Tidal Dynamics

The tidal component of the flow was estimated from the near bed vertically averaged AD2CP mean flow data shown in Figure 6. The short record length (~4 hours) of the time series does not allow resolution of the tidal signal using harmonic analysis. Instead, the mean velocities from a total of 80 stations (not shown here) located in the general area of the sampling site were utilized. These data were collected over a period of ~5 tidal cycles from April 12 to April 15, 2023, and the assumption is made that the spatial variability of the tidal signal in the area is not significant. Only depth-averaged velocity data from water depths > 4.5 m are used in the analysis to ensure that the tidal current estimates are not influenced by plume-associated flows. The harmonic analysis was carried out for the two horizontal flow components using only a major semi-diurnal (M2) and a diurnal (O1) constituent. In Figure 7 the reconstructed time series of the tidal flows are presented together with the velocity data used as input in the least-square analysis.

The amplitudes of the M2 and O1 constituents were 0.08 and 0.01 m s$^{-1}$ for the east (u) component and 0.01 and 0.03 m s$^{-1}$ for the north (v) component of the flow. The corresponding relative phases were 215.1 and 73.8 degrees for u and 67.5 and 251.5 degrees for v. The overall mean tidal flow values were −0.01 and 0.07 m s$^{-1}$ for u and v, respectively. As seen in Fig. 7, the arrival of the plume disrupted the tidal flow near the seabed significantly, creating deviations of approximately 0.10 m/s from the tidally expected flow.





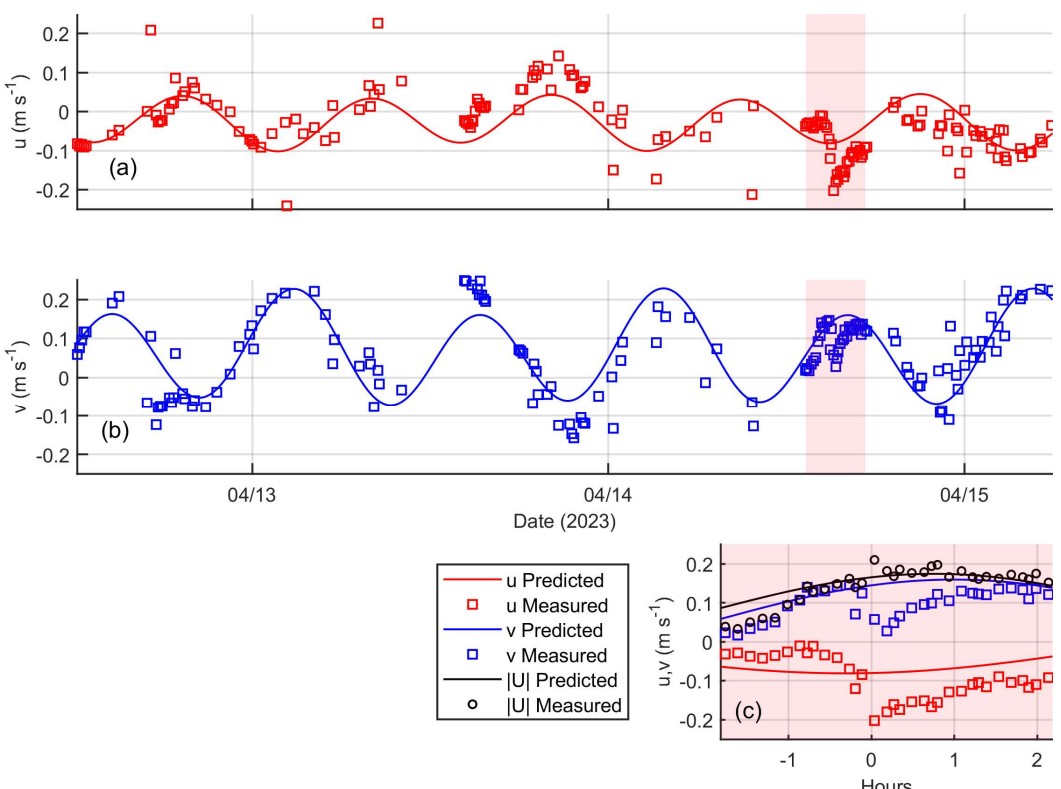

**Figure 7: Time series of bottom averaged (depths > 4.5 m) east (u) and north (v) velocity components from all stations occupied during the period April 12 - 15, 2023. The reconstructed tidal flow is shown as solid line and it was developed using the results of the harmonic analysis. The shaded area indicates the period station TS2 was occupied. Bottom insert shows a close up of the measured and predicted tidal flows with the time in hours relative to the time of plume front arrival.**


The tidal flow seems to increase in magnitude during the data collection period, achieving a maximum approximately +1 hours after the arrival of the front. In terms of total speed, the observed near bed mean flow does not seem to change its total magnitude, but instead the main impact is a change in direction from a NE to a WNW direction initially, changing to NW over time, against the direction of the propagation of the plume.

**3.6 Turbulence Dissipation**

Figure 8 shows turbulent kinetic energy (TKE) dissipation rate ($\varepsilon_k$) profiles estimated from the AD2CP vertical beam radial velocities using the second-order structure function (SF) method (see section 2.2). The gaps seen in the profiles at mid-water





levels are because of interference between two successive pulses and are a result of the shallow operating depth of the AD2CP. These data failed the MSPE (<5%) error criterion, as did some other data as shown in the figure below.

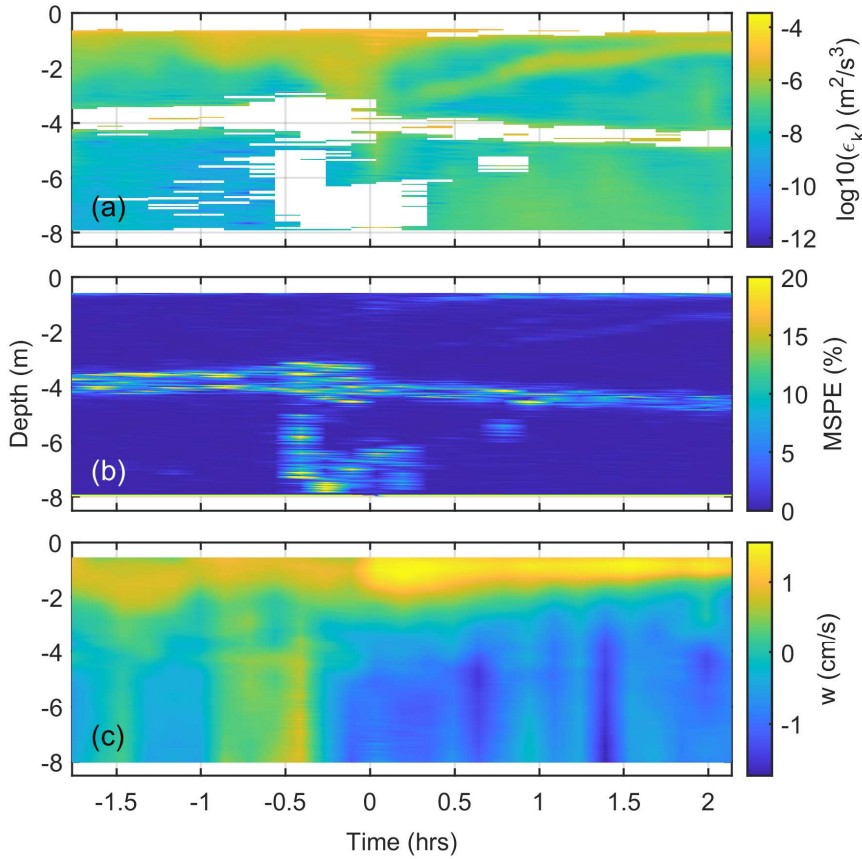

**Figure 8: Time series of vertical distribution of: (a) Dissipation rate ($\varepsilon_k$) estimated from the AD2CP 5th beam HR radial velocity profile using the structure function (SF) method (see section 2.2). The plot is created using 27 individual profiles corresponding to each one of the AD2CP 8.53 min ensemble. (b) Mean square percent error (MSPE) used to screen the data as per Zeiden et al. (2023). (c) Mean vertical velocity estimated after waves were removed using an EOF method (Zeiden et al., 2023).**


Prior to the front's passage, dissipation rates near the surface (< 2 m) are relatively high, of the order of $10^{-5}$ m$^2$ s$^{-3}$ and decrease with depth down to 4 m depth. Then in the lower water column (>4 m) dissipation rates remain relatively constant at approximately $10^{-9}$ m$^2$ s$^{-3}$. After the front's passage, the dissipation rates exhibit a more complex structure, especially near the surface (<4 m) with high rates near the surface as expected, reducing with depth to ~ $10^{-7}$ m$^2$ s$^{-3}$ and then increasing again

(secondary peak) before decreasing again toward the 4 m depth. The depth of the near surface secondary peak appears at ~ 3 m immediately after the passage of the front and reduces in depth with time reaching approximately 1 m about 2 hours after the





front's arrival. Near the bed (>6 m) dissipation rates are lower before the arrival of the front than after. This is attributed to benthic boundary layer processes and increased near bed total current speeds (see Fig. 7 bottom).

The vertical velocities estimated during the analysis of the 5th beam data are shown in Fig. 8c. These vertical velocities, although corrected for wave motion using the EOF method described in Zeiden et al. (2023), show a persistent positive (upward) flow close to the surface and predominantly negative (downward) flows at greater depths after the passage of the front. Nevertheless, there are times that positive flows appear although the magnitude is less than 1 cm s$^{-1}$.

At this juncture it should be noted that a comparative analysis of dissipation estimates from the MicroCTD and AD2CP showed qualitative agreement (Papageorgiou, 2023). The MicroCTD measurements are those from single profiles while the structure
function estimates integrate data over a longer period (8.53 min), thus any variability shown in the microstructure data would not be visible in the SF method derived values. Overall, the estimates from both measurement systems display similar trends in the vertical and are of similar order of magnitude; and the variability observed in the profiles (with a maximum discrepancy of one order of magnitude) is likely attributed to the differing averaging periods. This assertion is further supported by the observation that the discrepancies between the MicroCTD and structure function estimates are comparable in magnitude to the
variations observed among individual MicroCTD casts (Papageorgiou 2023).

## 4. Data Synthesis and Discussion

### 4.1 Plume Structure

As presented earlier, the T-S-density profile results presented in Fig. 5 reveal a water column that transitions from a single-step to a two-step density structure. This suggests the presence of a pre-existing plume (hereafter referred to as layer 1) from the
previous tidal cycle followed by the arrival of a newly discharged plume (layer 2) that propagates above it. The system transitions from a two-layer model (ambient waters and pre-existing fresher water mass) before the arrival of the front to a three-layer model afterward. Although various investigators have used an isohaline to delineate a plume (i.e., S=21 in Kastner et al., 2018, S=27 in McDonald and Geyer, 2004 etc.) the T-S diagram shown in Fig. 4 indicates that the plume depth is defined by temperature and salinity values >18.3°C and <32.7, respectively. These values indicate a plume depth of 2.9 m initially reducing to 1.9 m
toward the end of the data collection period. We quantitatively estimated the plume depth using the generalized equation of Arneborg et al. (2007), after modifying it for a three-layer structure (see Appendix A). The estimated depths for the pre-existing ($D_1$) and the new ($D_2$) plume are listed in Table 1 and provide each layer's vertical extend both before ($t<t_f$) and after ($t>t_f$) the front's arrival.

**Table 1: Depths of pre-existing ($D_1$) and newly arrived ($D_2$) plumes for the times corresponding to the different group profiles.**

|             | $t<t_f$ |     | $t_f$ | $t>t_f$ |     |
| ----------- | ------- | --- | ----- | ------- | --- |
| Layer Depth | A       | B   | C     | D       | E   |
| $D_1$ (m)   | 3.8     | 4.0 | 4.6   | 4.8     | 3.8 |
| $D_2$ (m)   | 0       | 0   | 2.6   | 2.5     | 1.7 |





Initially, the pre-existing plume (layer 1) has a depth ($D_1$) of 3.8 meters. The arrival of the newly discharged plume (layer 2) caused a depression in the interface between the pre-existing plume and the ambient waters by approximately 1 meter, though this depression gradually diminished over time. Over the course of the experiment, the depth of the newly discharged plume gradually decreased from 2.6 to 1.7 meters, indicating a reduction by a factor of 0.75 over a period of ~ 2 hours. The $D_1$ and $D_2$ depths presented above appear to correspond to the 34 and 32.5 isohalines, respectively.

Utilizing the depths in Table 1 and the density profiles shown in Fig. 5, the water density within each layer was estimated and a conceptual schematic of the water column during the experiment was created (Fig. 9). The schematic shows the average changes in density and depth of both the pre-existing and newly discharged plumes over time. The pre-existing plume maintained a depth close to 4 meters, with an average density of 1,023.8 kg m$^{-3}$ (S=33.1), overlying ambient ocean waters with a density of 1,024.8 kg m$^{-3}$ (S=34.3), while the new plume had an average density of 1020.7 kg m$^{-3}$ (S=29.3).

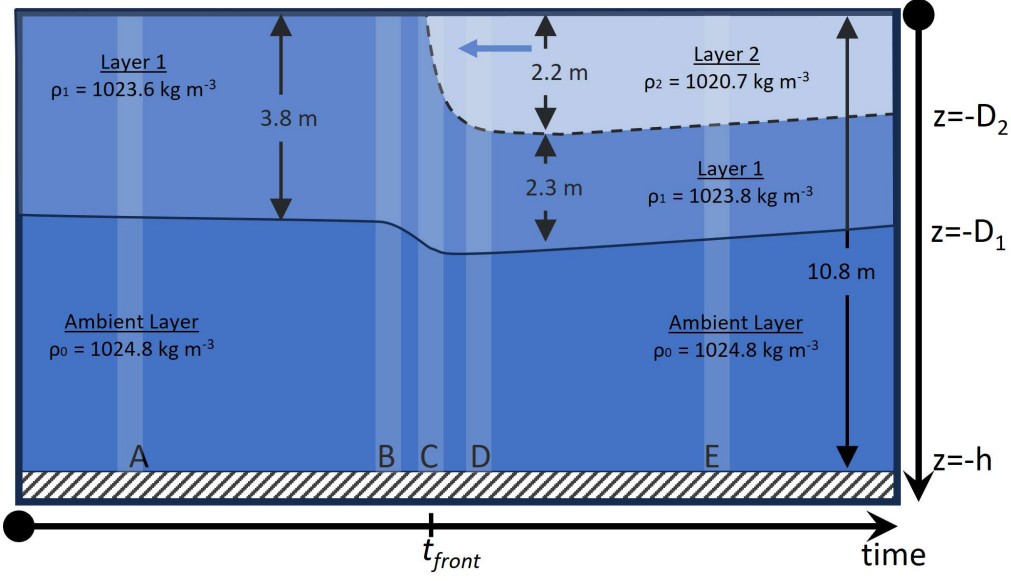

**Figure 9: Conceptual model describing the structure of the water column throughout the data collection period. Initially a 2-layer structure is present ($t < t_f$) where an upper layer of depth $D_1$ and density $\rho_1$, presumably from a previous plume, is over ambient ocean waters with density $\rho_0$. The front of a newly discharge plume of depth $D_2$ and density $\rho_2$ arrives at $t = t_f$, contributing to the creation of a 3-layer structure. The vertical shaded bands denote the time / relative location of the microCTD profile groups shown in Figure 3b.**

## 4.2 Plume Kinematics

In this section the kinematics of each layer (i.e., old, pre-existing and new plume) are examined using the AD2CP measured mean flows. Figure 10a shows vector diagrams of the horizontal flow (u, v) as measured by the AD2CP at the various bins below



tl

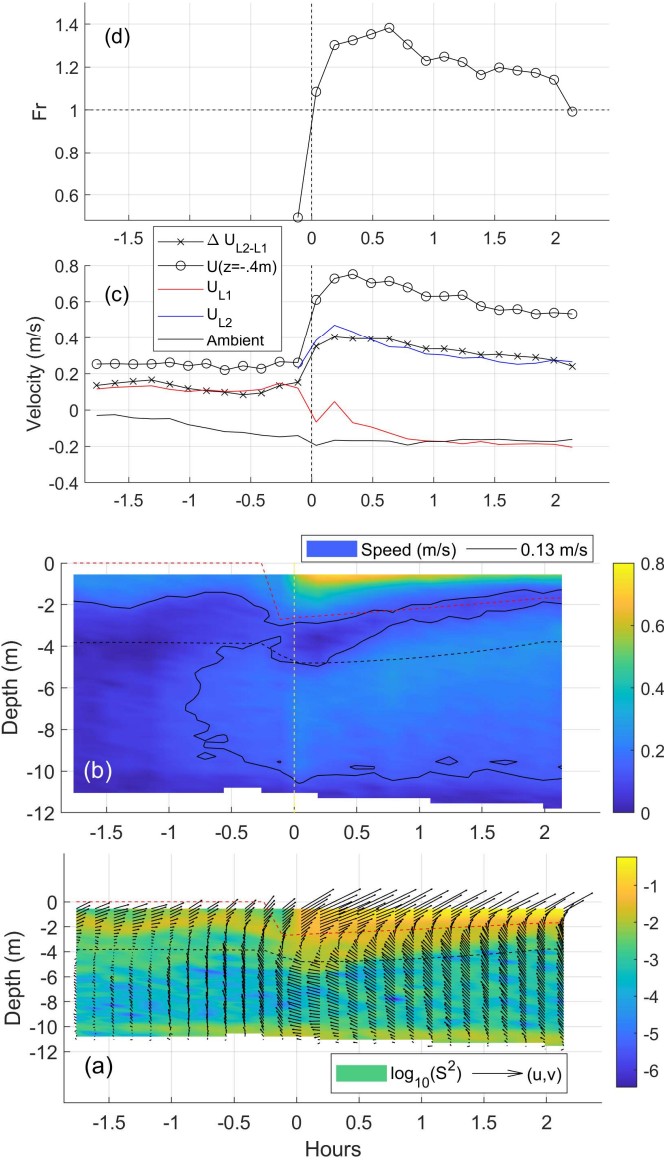

**Figure 10: Time series of vertical structure of the AD2CP measured horizontal flow and shear before (t<0 hours.) and after (t>0 hrs.) the passage of the front. (a) Vector diagram of horizontal velocity for various depths below the sea surface. The log of the velocity shear square ($S^2$) is shown in the background as a contour. (b) Contour plot of horizontal flow speed. The 0.12 m s$^{-1}$ contour is shown as it delineates times / elevations of relatively weak flows corresponding to times-elevations of flow deflections / reversal after the arrival of the front (t>0). (c) Time series of flow strength for various parts of the water column. Key: Ambient: layer-depth integrated flow speed for the ambient water mass ($z>D_2$); (d) Froude number variability inside the newly arrived plume. Key: $U_{L1}$ - depth integrated flow speed for layer representing the newly arrived plume ($z<D_1$); $U_{L2}$ - depth integrated flow speed for layer representing the pre-existing, old plume ($D_1<z<D_2$); U(z=0.4m) - surface velocity defined as the AD2CP recorded velocity from the bin close to the sea surface (0.40m); $U_{L1-L2}$ - Relative speed of the new plume estimated by subtracting the absolute velocity of the layer below it from that of the plume itself. Negative sign indicates flow toward the west.**



Figure 10c together with the absolute, layer-integrated, flow speeds of the ambient bottom layer waters ($z<D_1$, Ambient), the older, pre-existing plume layer (layer 1, $D_1<z<D_2$, $U_{L1}$) and the new plume (layer 2, $z>D_2$, $U_{L2}$) behind the front.

Prior to the arrival of the front, the surface layer (~2 m) speed is 0.22 m s⁻¹ towards the ENE. Further below (ambient water layer), the surface current decays to ~0.1 m s⁻¹ and reverses direction towards WNW. In these deeper (>4 m) waters there is no significant structure in the flow and the current velocity is small, increasing slightly over time. The vertical structure within layer 1 resembles that of an Ekman layer, suggesting that the movement of the older plume is driven by the wind that is toward the NE (Fig. 2). This was verified by comparing the de-tided flow measurements with a theoretical Ekman profile (Papageorgiou,

310    2023).

After the arrival of the new plume, the flow in layer 1 intensifies, reaching 0.75 m s⁻¹ near the sea surface some 17 minutes after the front's arrival. The current within layer 1 is toward the ENE and diminishes with depth, developing significant shear. Immediately behind the front, the flow within layer 1 diminishes with depth to a minimum velocity (<0.13 m/s) point that

corresponds to a change in direction further below toward the NW. This minimum velocity point moves to shallower waters over time and the direction below it shifts slightly from NW to NNW. Initially the minimum flow point is found at ~$D_1$. Toward the end of the data collection period, some 2 hours later, it is located at half the water depth (~$0.5D_1$). The location of this minimum flow area is graphically depicted by the 0.13 m/s current speed contour in Fig. 10b.

Time series of the layer-integrated flows are shown in Fig. 10c. The absolute speeds of layers 1 ($U_{L1}$) and 2 ($U_{L2}$) seem to be of similar magnitude (~0.20 m/s) at about 1.5 to 2 hours. after the arrival of the front. During the first 30 min after the arrival of the front the water speed of layer 1 is ~0.10 m/s, while that of layer 2 is ~0.40 m/s. It is worth noting that the flow deflection in layer 1 started some 15 min before the arrival of the front. The ambient layer is moving consistently toward the WNW before the arrival of the plume, with flow speeds increasing from a few cm/s to ~0.20 m/s at the time the front arrives. After that time the

water mass within this layer moves toward NW with the same speed.

### 4.3 Frontal Propagation

The absolute speed of the front was estimated from successive photographs collected from a drone (Fig. 11) as well as the upper bin of the AD2CP that was located at 0.40 m below the sea surface (see Fig. 10c). Both methods revealed a frontal propagation toward the ENE with an absolute front speed of 0.61 m s⁻¹. The water mass behind the front was moving along the same direction

as the front with absolute speeds of 0.72 and 0.75 m s⁻¹ some 8.5 and 17 min after the front passage, respectively. The layer-depth averaged velocities of the front and plume behind it, after subtracting the layer-averaged velocity of the layer underneath it (i.e., relative velocities, see Fig 10c) were 0.36 m/s and 0.40 m/s, respectively, suggesting an overtaking velocity of 0.04 m/s. The reduced gravity g' and the mean depth of the layer of the plume (layer 2, $D_2$) were found to be ~0.034 m s⁻² and 2.2 m respectively. Based on these values a frontal Froude number ($F_r = U_f/(g'D_2)^{1/2}$) of 1.32 is estimated. This value is between the theoretical

value of 1.42 expected for a freely propagating gravity current (von Karman, 1940) and the value of 1.19 suggested by Huppert and Simpson (1980). The flow remained supercritical ($F_r>1$), although diminishing over time for 2 hours after the front passed over the station. Using an average water depth of 11.5 m and the frontal speed of 0.36 m/s a frontal Reynolds number ($Re_f = $



$((U_f\,h)/\nu$ of 4.03 x $10^6$ is estimated, a value very similar to that found for the Merrimack River plume by Horner-Devine at al. (2013). The bulk-mixing coefficient (β, Simpson and Britter, 1979; 1980) was calculated to be 0.11.

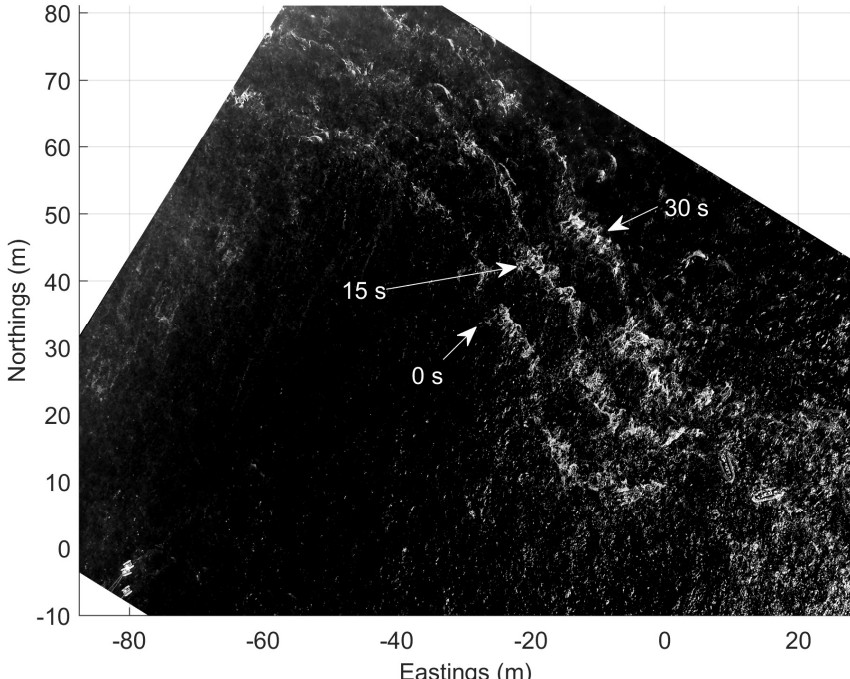

**Figure 11: Composite image created from the superimposition of three orthorectified aerial images of the front obtained at three different times, 15 s apart. Image rectification based on the drone GPS data while lens distortion was accounted for using an earlier version of the calibration package of Bouguet (2022).**


**Table 1. Vertically integrated and time averaged velocities (in cm/s) for each layer prior to ($t < t_f$) and after ($t > t_f$) the arrival of the front. For layer definitions see Fig. 9. Values derived from averaging in time the layer integrated values shown in Fig. 10 (top panel).**

|  | $t<t_f$ | | | $t>t_f$ | | |
|---|---|---|---|---|---|---|
| cm/s | u | v | \|U\| | u | v | \|U\| |
| Layer 2 - 1 | - | - | - | 32.5 | 1.7 | 32.6 |
| Layer 2 | - | - | - | 23.2 | 15.2 | 27.3 |
| Layer 1 | 8.4 | 5.8 | 10.3 | -9.3 | 13.5 | 16.4 |
| Ambient | -3.2 | 7.3 | 9.0 | -12.3 | 10.4 | 16.1 |





Although there are no direct observations of the spatial structure and spreading rate of the plume, the Sentinel image collected

the day after the experiment (see Fig. 1) provides a time snapshot of the plume shape and areal extent at that time. As in Pritchard and Huntley (2006) the area of the plume ($Ap(t_f)$=62 km$^2$) shown in the imagery was used to estimate the plume effective radius (=4.4 km) assuming a semicircular shape. Using the observed plume velocity of 0.33 m s$^{-1}$ (1.2 km/h) as a representative value of a linear spreading rate (see Pritchard and Huntley, 2006), then the change of the effective radius variability during the time elapsed between the front's passage from the sensors to the end of the data collection period (~2 hours) is ~ 6.7 km. Assuming

an instantaneous release of freshwater and conservation of mass, then the reduction of the plume depth over the 2 hours of observation is estimated to be ~ 0.43 m. This is an extreme value estimate assuming all river discharge occurred instantaneously at LW and there was not a continuous supply of fresh water. The shallowing of the plume observed is of the order of 0.60 (using the density profiles in Fig 5) which is greater than the 0.43 value but reasonable given the uncertainties of these estimates.

### 4.4 Mixing Processes

The density profiles (Fig. 5) from the periods corresponding to groups A to E were matched to corresponding mean velocity profiles (Fig. 6) to estimate the buoyancy frequency (N) and shear (S, where $S^2 = (\frac{\partial u}{\partial z})^2 + (\frac{\partial v}{\partial z})^2$ ) for each group. The process included interpolation of the flow data (dz=0.25 m) to the elevations of the buoyancy frequency profiles (dz~0.01 m), and application of a 32 point moving average (~ 0.32 m) prior to estimating gradients. Then these values were used to calculate the

gradient Richardson number ($Ri_g$=S$^2$/N$^2$) for each individual CTD profile (Fig. 12).

In group A, a slightly elevated buoyancy frequency is seen at ~2m depth that propagates slightly deeper in group B, just before the arrival of the plume. Near the front (group C) high buoyancy frequency values are found near the surface and although diminished, these high values extend deeper in the water column down to 6m. Immediately after the front's arrival (group D) although elevated buoyancy frequency is maintained up to 6 m, near surface values are smaller than those seen in group C. Later,

behind the front (group E) the maximum buoyancy frequency is observed near the surface with S reaching values of 0.1 s$^{-1}$. Similar variability is seen in shear with the strongest shear being observed in group E. A more detailed image of shear variability can be seen in Fig. 10 (bottom).

The gradient Richardson number estimates, although variable, do provide some insightful information. As explained earlier, these estimates are based on single CTD casts combined with averaged current profiles; as such the variability seen is to be

expected. Despite the variability near the bed, at depths >6 m (i.e., up to ~5 m above the seabed), $Ri_g$ values are consistently below the critical value of 0.25. This holds for all groups / periods of data collection, before and after the arrival of the front. The $Ri_g$ values near the surface, although variable for group A, are more consistent for group B. The estimates suggest mixing ($Ri_g$<0.25 or $\log_{10}(4 \cdot Ri_g)$<0) near the surface (~1 m for A, 0 to 2 m for B), something more clearly shown in group B, which coincides with the period of the highest observed wind shear stress of ~0.05 N/m$^2$. In depths 2 to 6 m below the sea surface, $Ri_g$

tends to be >0.25 suggesting that small shear present in that elevation is not effective in further mixing the pre-existing plume waters. The $Ri_g$ profiles for groups C, D, and E away from the near bed region seem to be between the 0.25 and 1.00 values (shaded region in Fig. 12) suggesting a fine balance between shear and buoyancy as both seem to increase in tandem with the arrival of the new plume.



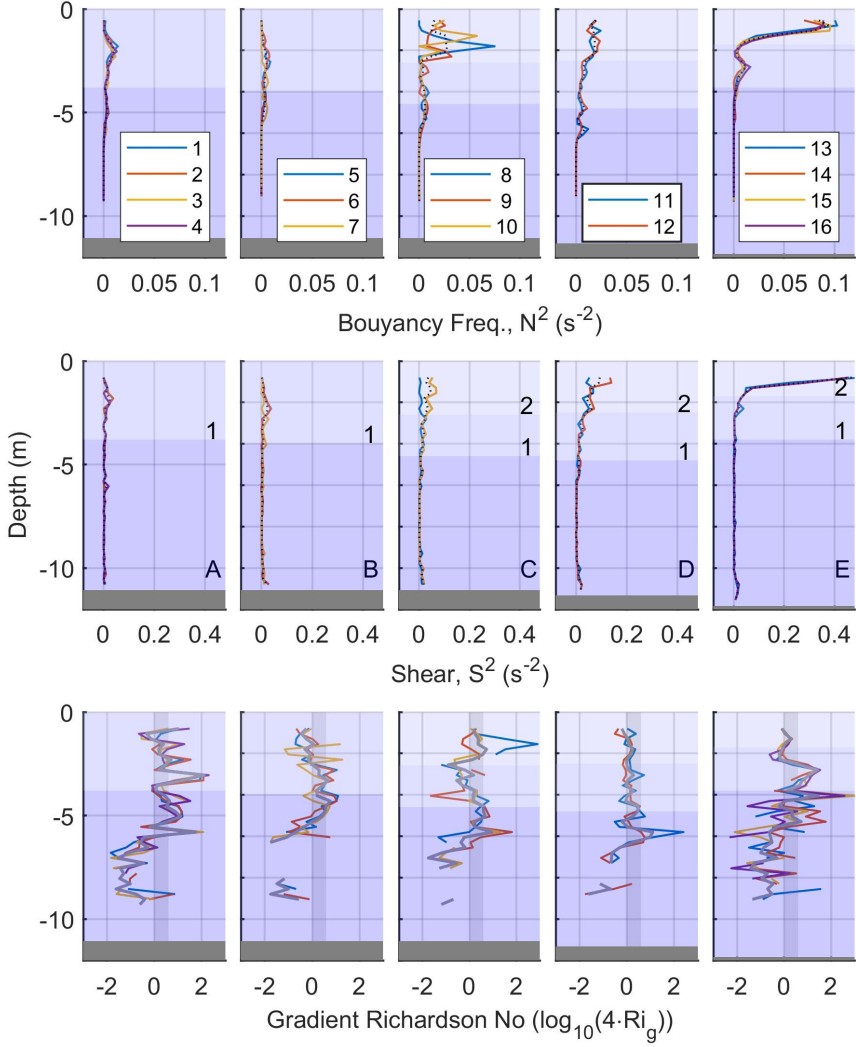

**Figure 12: Top: Buoyancy frequency (N²) profiles for each CTD profile shown in Figure 5 arranged by group (A to E). Middle: Corresponding shear (S²) profiles estimated using the nearest in time velocity profile (see Figure 6). Bottom: gradient Richardson numbers for each profile plotted as log10(4Ri$_g$); shaded area denotes the region of 0.25 <=Ri$_g$<=1. The different shades and numbers 1 and 2 represent the different water masses identified (1 – pre-existing plume, 2 – newly discharged plume).**

The mixing processes are further examined by looking at the "available overturn potential energy" (AOPE) in the water column, first presented by Dillon (1982) and later revisited by Smith (2020). The latter work presented an implementation method suitable for profile data with uneven vertical spacing, as is the case for our MicroCTD sensor. The method sorts the density values of the profile and estimates the relevant Thorpe scales ($L_T$). Following Smith (2020) the regions of the water column where the cumulative sum of $L_T$ becomes 0 are used to define parts of the water column where overturning occurs (i.e., turbulence patches). The size of each turbulence patch ($L_Q$) is estimated, and a constant Thorpe scale is assigned to each one of them. Using this



method we estimated the dissipation rate ($\varepsilon_p$) of turbulence potential energy (TPE) which together with the turbulence kinetic energy dissipation rate define mixing efficiency as (Smith, 2020):

$$\Gamma = \varepsilon_p / \varepsilon_k \tag{3}$$

$\Gamma$ is a parameter that essentially represents the amount of turbulence kinetic energy expended for mixing the water column in coastal stratified, quasi-steady shear flows; it is usually assumed that $\Gamma \approx 0.22$ (Peters, 1999; Kay and Jay, 2003; MacDonald and

Geyer, 2004; Osborn, 1980). However, Scotti (2015) showed that for isotropic overturn this value is 2. On the other hand, Burchard and Hofmeister (2008) in their work on mixing and stratification in estuaries and coastal seas using Simpson's (1981) concept of potential energy anomaly $\phi$ used a bulk mixing efficiency $\Gamma = 0.04$, while Simpson et al. (1990) used a value of 0.003. This uncertainty on the value of the mixing efficiency coefficient provides some uncertainty in estimating density diffusivity ($K_\rho$) that is usually expressed as:

$$K_\rho = \Gamma \varepsilon_\kappa / N^2 \tag{4}$$

Smith (2020) suggested that the ratio the Thorpe scale ($L_T$). representing overturn, over the size of a turbulence patch ($L_Q$) where the overturn takes place can provide an insight into the mixing mechanism (i.e., shear-flow vs. density inversion driven). This was called the L-ratio, and it was empirically related to the slope ratio (m-ratio) defined as the slope of a linear fit of the raw data over the slope of a linear fit to the sorted data (equivalent linear stratification, Mater et al., 2015). An m-ratio of -1 represents

what Smith (2020) defines as a "young" patch corresponding to conditions of purely buoyancy induced turbulence while a value of +1 represents mixing by pure shear-driven turbulence. Smith (2020) suggested this parameter as a potential indicator of the relevant contribution of the kinetic and potential components in total turbulence energy and dissipation.

The dissipation rate of turbulence potential energy (TPE) is estimated as:

$$\varepsilon_p = \frac{(0.55 + 0.45 \, m/\hat{m})}{\alpha} L_T^2 N_A^3 \tag{5}$$

where $\varepsilon = 4$, $m/\hat{m}$ is the m-ratio described earlier, and $N_A$ is the equivalent buoyancy frequency derived from the equivalent linear stratification defined as (Smith, 2020):

$$N_A = \frac{\sqrt[2]{2 \, AOPE}}{L_T} \tag{6}$$

AOPE is the change in potential energy before and after sorting (see equation (9) in Smith, 2020).

The results of this analysis are shown in Figs. 12 and 13. First the vertical distribution of the Thorpe overturning scales for each

one of the profiles, arranged by group, is shown in Fig. 13 (top). These plots appear to be more informative than the $Ri_g$ plots (Fig. 12) discussed earlier. The larger overturning scales are found near the bed with values ranging from 0.20 to 0.50 m. The smaller (~0.20 m) near bed scales are found at group A, slightly increasing at group B, prior to the arrival of the plume and attaining their largest values (~0.50 m) during and after the arrival of the new plume (groups C to E). Similarly high overturning Thorpe scales are estimated near the surface, limited to depths <1.5 m in group A profiles. This layer of high $L_T$ scales deepens

over time to 2.5 and 4.0 m for group B and C profiles, respectively. After the arrival of the new plume, there are no or very limited $L_T$ estimates, as the sorting of the density profiles did not identify any overturns or the ones estimated were of the order of the vertical resolution of the CTD profiles (~0.01m). The latter values are not considered significant as they can be artifacts of instrument noise. In groups A to C no significant overturning is revealed in the region between the surface and bottom layers.





The same results are shown in Fig. 13 (middle) where the extent of the regions of overturning (i.e., turbulence patches) are
shown.

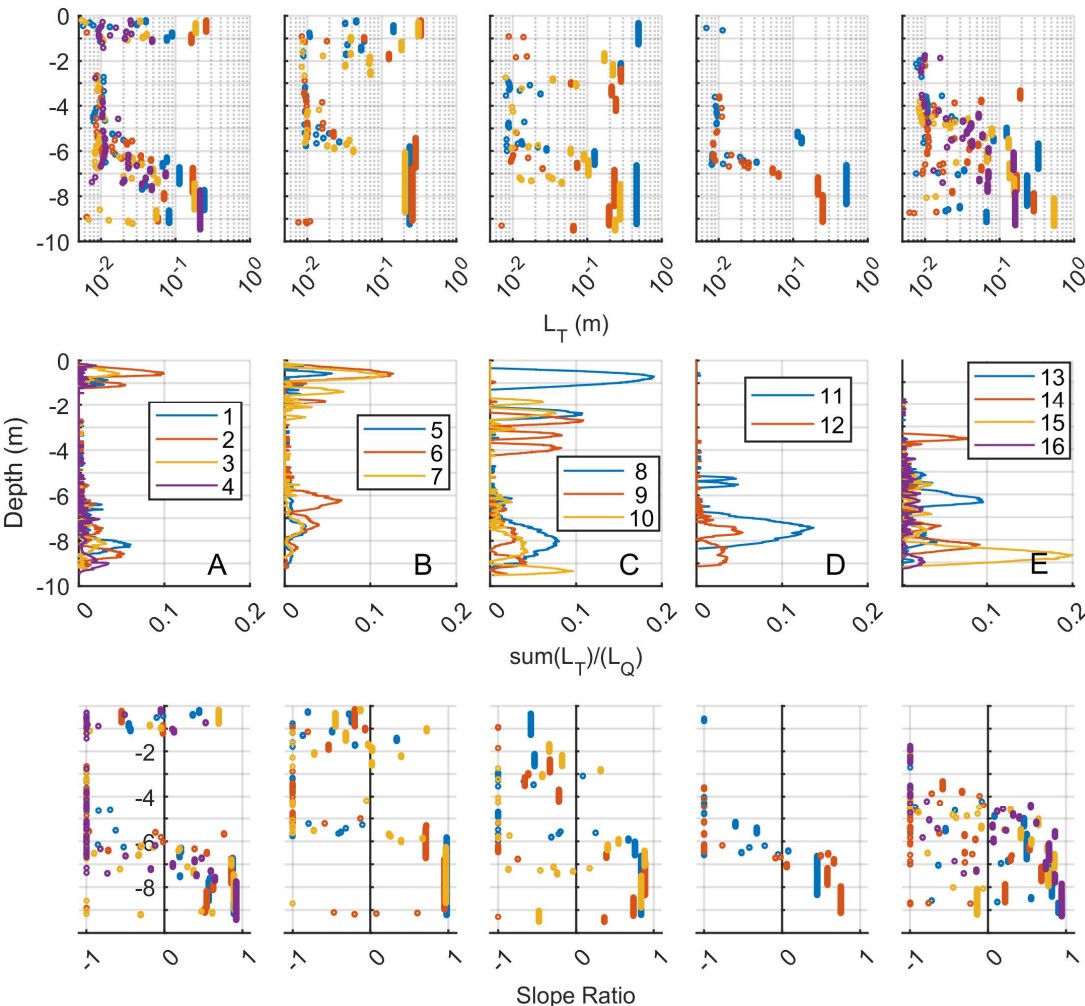

**Figure 13: Results of the Smith (2020) method for mixing using the water column turbulent potential energy. Top: Thorpe scales ($L_T$) estimated from sorting individual density profiles (see Figure 5). Middle: Vertical extend of turbulence patches estimated from summing the estimated Thorpe displacements; the values shown are the sums normalized by the scale (length) of each individual turbulence patch. Bottom: Vertical distribution of the slope ratio indicating the potential mixing mechanism (see text for details). Each column corresponds to a different profile group (A to E) representing different times in relation to the time the plume front passed over the station.**

The slope ratio profiles (Fig. 13, bottom) suggest that the near bed overturning regions, described above, coincide with regions of ratios close to +1 suggesting shear flows being the main driving force. Near the sea surface the slope ratio values show a





scatter from -1 to +1 for group A. The scatter reduces to the range -1 to 0 in group B, suggesting a diminishing contribution of

shear flows. This narrow range continues to group C where slopes are between -1 and -0.5.

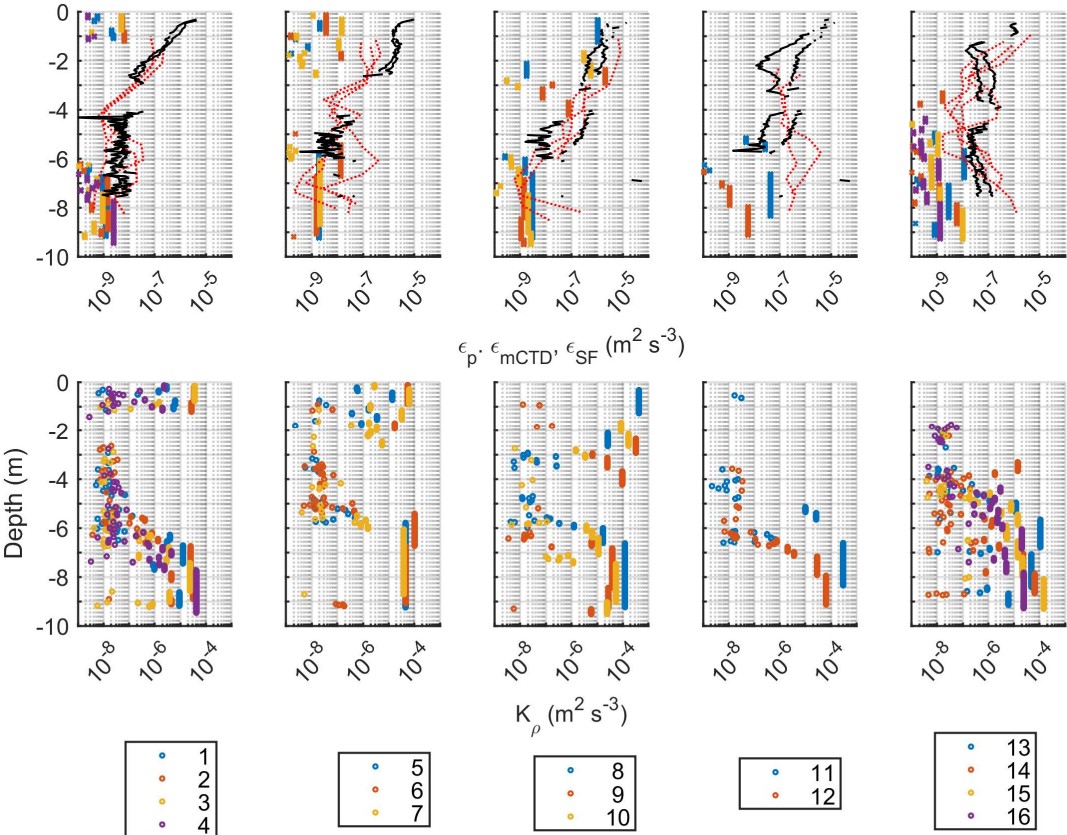

**Figure 14: Results of the Smith (2020) method for mixing using the water column turbulent potential energy. Top: Profiles of dissipation rate of turbulent potential energy ($\varepsilon_\rho$) and TKE from the microstructure (MicroCTD) profiler (red dashed lines) and the AD2CP (black lines) after applying the Structure Function (SF) method (see Figure 8a). Bottom: Density diffusivity ($K_\rho$) estimated using the Smith (2020) method (see text for details). Each column corresponds to a different profile group (A to E) representing different times in relation to the time the plume front passed over the station.**

Figure 14 shows the vertical distribution of the estimated TPE dissipation rates (equation 5, Fig. 14 top) and density diffusion coefficient (equation 4, Fig. 14 bottom). For comparison, the TKE dissipation rate profiles from the AD2CP (SF, see Fig. 8) and the microstructure on the MicroCTD are also shown on the same figure as black solid and red dotted lines, respectively. The

main points from these plots are that: (i) the TKE dissipation rates ($\varepsilon_k$) from the instantaneous microstructure CTD profiles and the averaged AD2CP HR radial velocities show a qualitative agreement.; (ii) near the bed, in the regions where the slope ratio was found to be ~ +1, the TKE and TPE dissipation rates appear to be of the same order, making the argument of a mixing efficiency coefficient ($\Gamma$) with values between 0.2 and 1 plausible; (iii) near the surface in group A to C profiles, $\varepsilon_\rho \ll \varepsilon_k$, by at





least 2 orders of magnitude, suggesting Γ values <0.01 as suggested by Burchard and Hofmeister (2008); finally (iv) within the
new plume, although high TKE dissipation rates are estimated by both the SF method and the microstructure, the overturning
analysis reveals no vertical mixing. This discrepancy within the new plume could potentially be attributed to anisotropic
turbulence, or turbulence advection, something not resolvable by the data available in this study. The high velocity shear found
inside the new plume suggests that straining could be a potential mechanism for maintaining the observed stratification inside
the plume, which to our knowledge has not been assessed for plume interiors. As additional information on the structure of the
water column, we present the acoustic backscatter from the AD2CP transducers. It is known that acoustic returns include
scattering by both turbulent salinity microstructure and particles in suspension (Lavey et al, 2013). Our data lack sound spectral
characteristics to allow for separation of the two backscattering sources and in addition the acoustic frequency of 1MHz falls in
the region where both salinity microstructure and sediment-induced backscatter are equally important (see Fig. 2 in Lavey et al.,
2013). However, we know that near bed resuspension levels are related to higher turbulence levels in the benthic boundary layer,
while on the surface, plumes tend to carry out river borne particulates that could contribute to higher backscatter. In Fig. 15 the
intensity of the return acoustic signal from the HR 5th beam (vertical resolution 0.04 m, sampling frequency 8 Hz) and transducer
1 of the broadband array of the sensor (vertical resolution 0.25 m, sampling frequency 1 Hz) are presented after they were
averaged using a 16s window. The HR return (Fig. 15a) does not extend beyond 7.5m depth while the BB transducer 1 (Fig. 15b)
extends all the way to the seabed. Both data have been corrected for geometric spreading and attenuation due to water absorption.

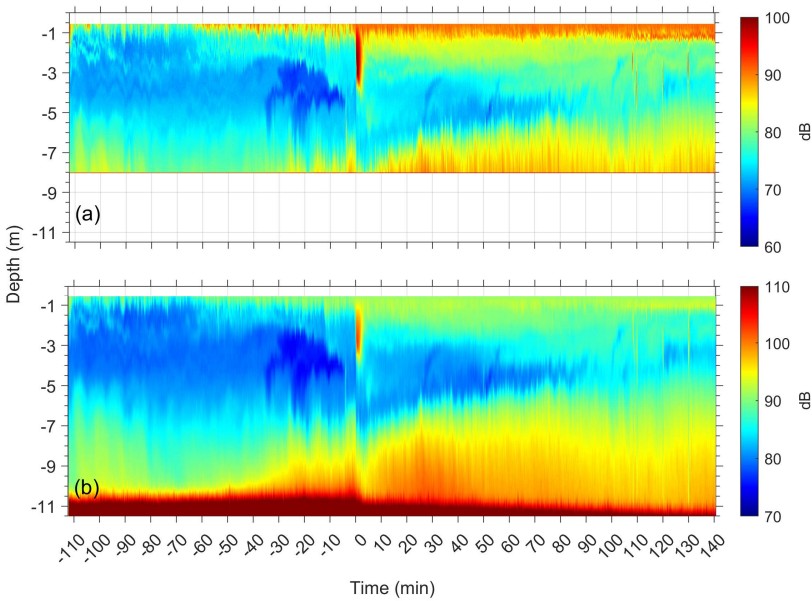

**Figure 15: Time series of acoustic backscatter profiles corrected for geometric spreading and water attenuation. (a) From the HR 5th beam record with a vertical resolution of 4cm, extending to ~7.5 m depth; (b) from one of the Broad Band transducers (beam 1) extending all the way to the sea bed (vertical resolution 25cm). Both records are 16 sec averages of individual pings collected at 8 Hz and 1 Hz for the HR and BB transducers, respectively. Units in dB; time in minutes relative to the passage of the front.**




The acoustic intensity map in Fig. 15 shows three major characteristics: (1) increased acoustic backscatter near the sea bed that is confined at depths >6 m prior to the arrival of the plume and then after the front arrival expanding to shallower depths (~4m) toward the end of the time series; (2) increased intensity near the sea surface after the arrival of the plume that could be due to river borne particle convergence near the front and / or increased turbulence; (3) the development of a lower backscatter region

455    in mid-waters with a vertical extent that reduces over time after the arrival of the plume, suggesting a merging of the top and bottom layers mainly due to expansion of the bottom layer. It is in this region of lower backscatter intensity, prior to the arrival of the plume, that evidence of internal waves (IW) with periods varying between 5 and 9 minutes is present. These IWs correspond to the times and depths (~2 m) of the IWs identified in the lower sensors of the thermistor array (see Fig. 3b).

Finally, as a summary, the vertically integrated layer estimates of TKE and TPE dissipation rates, along with density diffusivity,

460    are shown in Fig. 16 for the different groups / times before (A, B), during (C) and after (D, E) the passage of the front. The same values are listed in Table 3.

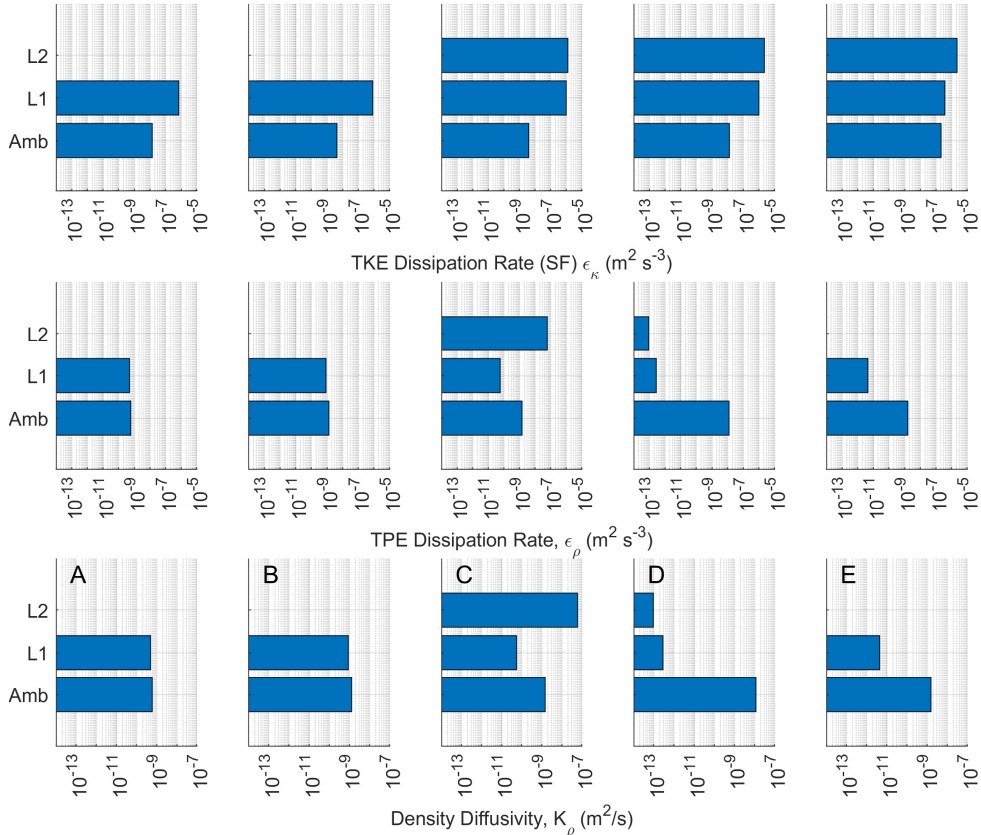

**Figure 16: Layer, vertically-integrated values of TKE (top) and TPE (middle) dissipation rates and density diffusivity (bottom). Amb: Ambient bottom waters; L1: pre-existing lighter water layer; L2: newly arriving plume.**





**Table 3. Layer integrated estimates of dissipation rate of turbulent potential energy ($\varepsilon_p$) and density diffusivity ($K_\rho$) for the different regions before (A, B), at (C), and after (D, E) the plume front passage.**

| | | A | B | C | D | E |
|---|---|---|---|---|---|---|
| $\varepsilon_k$ (m²/s³) | Layer 2 | | | $1.28 \times 10^{-6}$ | $2.19 \times 10^{-6}$ | $2.27 \times 10^{-6}$ |
| | Layer 1 | $6.88 \times 10^{-7}$ | $8.05 \times 10^{-7}$ | $9.51 \times 10^{-7}$ | $9.72 \times 10^{-7}$ | $3.74 \times 10^{-7}$ |
| | Ambient | $1.33 \times 10^{-8}$ | $4.25 \times 10^{-9}$ | $3.93 \times 10^{-9}$ | $1.27 \times 10^{-8}$ | $2.07 \times 10^{-7}$ |
| $\varepsilon_p$ (m²/s³) | Layer 2 | - | - | $5.94 \times 10^{-8}$ | $9.15 \times 10^{-14}$ | ~0 |
| | Layer 1 | $4.93 \times 10^{-10}$ | $8.64 \times 10^{-9}$ | $5.59 \times 10^{-11}$ | $2.81 \times 10^{-13}$ | $4.49 \times 10^{-12}$ |
| | Ambient | $5.09 \times 10^{-10}$ | $1.29 \times 10^{-9}$ | $1.44 \times 10^{-9}$ | $1.20 \times 10^{-8}$ | $1.56 \times 10^{-9}$ |
| $K_\rho$ (m²/s) | Layer 2 | - | - | $1.75 \times 10^{-5}$ | $1.78 \times 10^{-10}$ | 0 |
| | Layer 1 | $3.09 \times 10^{-6}$ | $6.35 \times 10^{-6}$ | $7.84 \times 10^{-7}$ | $8.08 \times 10^{-10}$ | $3.82 \times 10^{-8}$ |
| | Ambient | $1.02 \times 10^{-5}$ | $2.30 \times 10^{-5}$ | $2.62 \times 10^{-5}$ | $8.30 \times 10^{-5}$ | $2.24 \times 10^{-5}$ |

It is notable that within the plume, density diffusivity is highest O(-5) during the passage of the front (group C) and quickly reduces to O(-10) soon after (group D) to diminishing some 1-2 hours later (group E). This suggests reduction of plume thickness due to spreading as being the most dominant mixing mechanism. The introduction of the new plume also leads to a reduction in density diffusivity from O(-5) prior to the arrival of the plume to O(-8) to O(-10) after the new plume arrival. Near the bed density diffusivity appears to remain constant ~$3 \times 10^{-5}$ m² s⁻¹.

## 5. Conclusions and Summary

Comprehensive data highlighting the kinematics and mixing processes within a tidal river plume and its interaction with a pre-existing plume and the ambient waters were presented. The flows and mixing processes some 2 hours before and after the passage of a front at a location with a total depth of 11.5 m were analyzed. Prior to the arrival of the new plume, a pre-existing plume extending some 4 m below the sea surface was present. The water density of the pre-existing front was 1,023.6 kg m⁻³ while the underlaying ambient waters had a density of 1,024.8 kg/m³. The water density of the newly arrived front was 1,020.7 kg/m³ and its depth was 2.6 m, diminishing over time mainly due to radial spreading.

The relative propagation speed of the front associated with the newly discharged plume was 0.36 m/s while behind the front the propagation speed was 0.40 m/s. A frontal Froude number of 1.32 was estimated suggesting that the new plume was propagating as a gravity current.

Mixing processes were examined using the available overturn potential energy in the water column, as presented by Smith (2020). This analysis revealed mixing in the bottom boundary layer dominated by shear, while near the sea surface, prior to the arrival of the new plume mixing was dominated by a mixture of overturning and wind-induced shear flows. However, within the gravity current, and further from the frontal area, the mixing efficiency of the shear-induced turbulence was very small, despite the high TKE dissipation rates measured in that region.

Traces of IWs were identified in both the temperature time-series and acoustic imaging of the water column, particularly in the pre-existing plume, before the arrival of the new plume and at water depths characterized by high (>1) $Ri_g$ values.



**Data Availability**

The data presented in this manuscript are available as MATLAB (*.mat) files (Papageorgiou et al., 2025). For a description and access visit Zenodo ( https://doi.org/10.5281/zenodo.14687082).

**Code Availability**

The data were analyzed using standard analyses techniques implemented in MATLAB and modifications of the codes listed below.

The Structure Function analysis was carried out using the Zeiden et al. (2023) method and their codes are publicly available at https://github.com/SASlabgroup/SWIFT-codes.

The Smith (2020) analysis for mixing using the water column turbulent potential energy was done using a modified version of the MATLAB software code published in Smith (2020) online supplemental materials available at https://journals.ametsoc.org/view/journals/atot/37/1/jtech-d-18-0234.1.xml?tab_body=supplementary-materials.

**Author Contributions**

GV, AY and DF conceived the study, designed the experiments and participated in data collection. GV and CP did the data processing and analysis and wrote the manuscript with contributions from all co-authors. AY and DF handled the data processing for the microstructure and microCTD instrument. GV and CP developed the RoboCat and thermistor array system used in this study. GV, AY and DF acquired funding and managed the project. All authors reviewed and edited the draft
version.

**Competing interests**

The authors declare that they have no conflicts of interest.

**Acknowledgments**

The authors would like to thank the captain and crew of the R/V *Savannah* for their dedication and help during the data
collection period. Some of the technology developed for this project would have not been possible without the help Dr. D. Cahl and Mr. M.C. Wescott. Dr. Cahl also assisted in data collection as did several undergraduate and graduate students from the University of South Carolina and Coastal Carolina University. This material is based upon work supported by the U.S. National Science Foundation (NSF) under Grant Nos 2148479 and 2148480. George Voulgaris' contribution was partially carried out while at NSF as part of his approved Independent Research/Development (IR/D) plan. Any opinions, findings, and
conclusions expressed in this material are those of the authors and do not necessarily reflect the views of the NSF.





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



**APPENDIX A**

The plume depth was quantitatively estimated using the generalized equation of Arneborg et al. (2007), which was originally

developed to estimate the thickness (D) of a bottom gravity current:

$$D = \frac{2 \int_{-h}^{0} \frac{\rho(z) - \rho_0}{\rho_0} z \, dz}{\int_{-h}^{0} \frac{\rho(z) - \rho_0}{\rho_0} dz} \qquad\qquad\qquad\qquad A.1$$

where, z is the elevation below sea surface, *h* is the total depth of the water column, $\rho(z)$ is the density at elevation z and $\rho_o$ is the reference ambient density (1,024.8 kg/m$^3$ in this study).

Given our observations that a two-layer structure is developed into a three-layer structure we modified equation (3) so that we

can track the depth of the pre-existing ($D_1$) and the new ($D_2$) plume reflecting the system's state both before ($t<t_f$) and after ($t>t_f$) the front's arrival at time $t_f$. The modified equations are:

$$D_1(t) = \frac{2 \int_{-h}^{0} \frac{\rho(z,t) - \rho_0}{\rho_0} z \, dz}{\int_{-h}^{0} \frac{\rho(z,t) - \rho_0}{\rho_0} dz} \ , \ \text{for } t < t_f \qquad\qquad\qquad A.2$$

$$D_2(t) = 0, \qquad \text{for } t < t_f \qquad\qquad\qquad\qquad A.3$$

$$D_2(t) = \frac{2 \int_{-<D_1(t<t_f)>}^{0} \frac{\rho(z,t) - \rho(-D_1,t)}{\rho(-D_1,t)} z \, dz}{\int_{-<D_1(t<t_f)>}^{0} \frac{\rho(z,t) - \rho(-D_1,t)}{\rho(-D_1,t)} dz}, \quad t \geq t_f \qquad\qquad A.4$$

$$D_1(t) = \frac{2 \int_{-h}^{-D_2(t)} \frac{\rho(z,t) - \rho_0}{\rho_0} (z - D_2(t)) \, dz}{\int_{-h}^{-D_2(t)} \frac{\rho(z,t) - \rho_0}{\rho_0} dz} + D_2(t), \quad t \geq t_f \qquad A.5$$