# Peer review of "Flow Structure and Mixing Near a Small River Plume Front: Winyah Bay, SC, USA"

_EGUsphere, 2025_

## Referee Comment (RC1)

This is an excellent data set with great potential as a valuable contribution to the river plume community. My general comment is there is a lot going on in this manuscript: the figures are busy, and there are many redundancies. I find myself often losing track of what is important and what is extra "fluff". It think the results, discussion, and associated figures can be streamlined and the story made clearer. I am generally fine with the analysis, but some relatively significant restructuring and rewriting is needed. I tried to specify my individual concerns below.

-Preston Spicer, PNNL

**Major Comments:**

1. Introduction: there are not clear knowledge gaps outlined, so it is not clear if this data set and study are novel. Other folks have studied turbulence and mixing around plume fronts...what is new here? Make it clearer.
2. The tidal dynamics section (3.5) needs more clarifying information.
3. Restructuring associated with explaining results is needed. There are simply too many figures / too much text that are often redundant. It is very easy to get lost when reading from the amount of data.
4. Similarly, there are lots of turbulence/mixing parameters presented but it is not clear if they are all contributing to the story. Mixing is probably the most important part to this paper, but I feel the authors' do not guide the reader properly through their train of thought. Some variables appear, are not really discussed, or simply don't matter. I think there is also a major lack of discussion on dynamics/processes with too much focus on describing patterns in turbulence. Look through the figures and text and only present what is needed for your story. Repackage it.
5. The elevated currents and turbulence at depth after plume front passage is very interesting, but I don't think the dynamics associated with this are ever addressed. Why do the currents speed up so much at depth after the front passes? Is bottom boundary mixing (Spicer et al., 2021 JPO; Whitney et al., 2024 JGR) important? This has **not been observed** in many systems and would be very impactful/novel to quantify here.

References:

Spicer et al., 2021: The Effect of Bottom-Generated Tidal Mixing on Tidally Pulsed River Plumes, Journal of Physical Oceanography, https://doi.org/10.1175/JPO-D-20-0228.1

Whitney et al., 2024: Mixing of the Connecticut River Plume During Ambient Flood Tides: Spatial Heterogeneity and Contributions of Bottom-Generated and Interfacial Mixing, JGR: Oceans, https://doi.org/10.1029/2023JC020423

**Minor Comments (by line number):**

53          Says "near surface mixing" here but bottom layer mixing is mentioned in the abstract.

Fig. 1      Do you have bathymetry contours that could be added to Fig. 1a? This would be very helpful to the reader.... particularly if bottom boundary mixing is discussed in the paper.

| | |
|---|---|
| 99 | How far away, exactly, was the vessel? |
| 108-121 | We did a very similar upriser-mode MicroCTD sampling scheme in the Merrimack River plume (Spicer et al., 2022): https://agupubs.onlinelibrary.wiley.com/doi/full/10.1029/2022GL099633. It may be useful for you to cross-reference our paper for differences in turbulence estimation parameters. Also, did you take multiple profiles are average them together? Or just use singular profiles? |
| 144-146 | I do not understand this logic. Why such a difference in discharge estimates? How far away is the USGS gauge? |
| 146 | Why not show the tides? Seems important for a plume front that forms with the tide. |
| 148 | Is this low water in reference to low water at the NOAA gauge or at Winyah Bay? |
| 151 | Can you use nearby NOAA wave buoys to back this up? |
| Fig. 2 | I think it would be good to show river discharge and tidal elevations as well so we can see how the sampling day conditions fall in to the broader period. |
| 173 | You should mention the number of casts and when they were taken (i.e., a group before the front, a group in the front, a group after the front) in the methods section above. |
| 180 | Label the groups in Fig. 3b. Perhaps color the vertical lines marking the profiles the same colors as those in Fig. 4. |
| Fig. 4, 5 | Why is group A warmer and fresher than group B? |
| Fig. 6 | Why not make contour plots with time on x-axis, depth on y-axis, and velocities colored? If the ADCP was in a fixed position, this makes the most sense. That would explicitly show time variation and be a little easier to interpret/align with Fig. 3 time variation. |
| | After reading through this section more, I feel strongly this should be a contour plot. Although the current profiles show an obvious increase in plume layer velocities, it is very hard to distinguish what velocities are positive and negative here particularly in the sub-plume layers. Contour plots with a red-blue colormap would be ideal and helpful. |
| 206-208 | Mark the front arrival time on the velocity plot/contours. |
| 227 | A map of where these stations are located should be given in an appendix or supplemental material. What are these 80 stations? 80 ADCPs set up at the same time? This seems excessive and unlikely. More information is needed. |
| 230 | Why 4.5 m? It would appear the plume influences currents below that depth in Fig. 6. |
| 231 | Why only these two harmonics? |

| | |
|---|---|
| Fig. 7 | The velocity jumps evident at hour 0 in the inset would indicate there is plume influence here and the 4.5 m threshold is not the best to use. You refer to these jumps being a result of plume influence near bottom. Was the total depth at T2 ever mentioned? If the average includes plume velocities near 4.5 m, then I wouldn't call this near bottom. What do these scatter points look like if the threshold is deepened to 5 or 6 m? How deep is the water column at all the stations used here? Does it vary? By how much? |
| 242 | If the current changes direction exactly when the plume front passes, I don't think you can call this the tidal current...going back to my comments above. This is some combination of tide and plume. At all the other stations used to predict the tidal current, are they within or outside the plume? Why do we not see these jumps elsewhere? What accounts for the outliers in Fig. 7a and b which deviate significantly from the predicted tidal curve? |
| Fig. 8 | You probably don't need the TKE dissipation rate colorbar going as low as 10^-12.... this is essentially unimportant noise. I would constrain to 10^-10 or even 10^-8 to better emphasize where the strong turbulence is. Also, the MSPE panel is probably not necessary as long as you explain in the text why data is missing, which you have. |
| | Going back to my comment on making contours of the current velocities, you should add them to this figure (after removing MSPE) and moving this to Fig. 6. Then you can describe currents and associated turbulence together. Alternatively, you can replace MSPE with contours of shear squared ($S^2 = du/dz^2 + dv/dz^2$) to check how vertical velocity shear aligns with turbulence. |
| | Lastly, make the colormap of the vertical velocities red-blue so we can easily identify what is upward and downward moving velocity. |
| 263-270 | If this is true, please add an appendix or supplemental material to confirm to the reader. Typically, MicroCTD estimates are more reliable than those from an ADCP. It is strange to throw out those profiles here. |
| | Further, the thesis being referred to sharing this information has a link that does not work. |
| Fig. 10 | Panel (a) should be at the top with succeeding panels beneath. I see now the shear squared which is good. I still think shear should be shown with turbulence, as they are linked. I don't think the current vectors are helpful on this map (as it is not oriented with the cardinal directions) and are better shown as red-blue colormaps as I described previously. Also, showing u and v here again is redundant to Fig. 6. Combine like-things rather than describe them twice in two different places. |
| 305-325 | Haven't current speeds and direction already been discussed? This sections is quite redundant and should be integrated into the results where velocities were already discussed. |

| 330-332 | This delta U_L1-L2 variable is not clear. Is it the difference in U_L1 and U_L2....it doesn't seem that way considering it remains well above 0 in Fig. 10. |
|---|---|
| 337 | Can you describe why the frontal Reynolds number is being calculated. What is the relevance to your story. |
| Table 1 | If this information is presented in Fig. 10, then it seems redundant to include in a table as well. |
| 357 | Equation for buoyancy frequency? |
| 358 | Interpolation should be from the finer resolution to coarser: i.e., $N^2$ interpolated onto $S^2$ depth vectors. |
| Fig. 12 | You already showed $S^2$ in Fig. 10. These are more unnecessary redundancies which take up figure space. |
| 370-378 | Ri can be greater than 0.25 and still have intense mixing. This is particularly true for convective instability driven mixing, where often Ri >1, and we would expect to be happening in the front (see Ivey et al., 2020: https://agupubs.onlinelibrary.wiley.com/doi/full/10.1029/2020GL089455). I think your Ri profiles need to be smoothed more to get helpful info. By interpolating onto the coarser ADCP depth vectors rather than the finer CTD depths, this should help. |
| Eq. 3-5 | Kind of an odd order to presenting these equations. Equation 3 uses Equation 5, so it makes more sense to have Equation 5 come before 3. |
| Fig. 13 | This is too much to look at. Can you average by group and not show every single profile? This would be helpful here and in other figures. Where are the depth labels on the bottom panel? |
| 420-425 | I am interested how your cleaned up Ri profiles compare to this metric. Ri can indicate where convective or shear mixing is expected as well. It would be valuable to present the two parameters together and discuss alignment or lack thereof. |
| 420-430 | Again....is 10 m "near-bed"? I don't think total depth was ever mentioned. |
| Fig. 14 | ADCP TKE dissipation estimates were already presented way before this in Fig. 8....where it makes sense to show comparisons with the other TKE dissipation estimation methods. Combine or remove to eliminate more redundancy. |
| 433 | Elaborate on why these mixing efficiency values are notable. This would imply convective mixing, correct? |
| 439-458 | This whole backscatter section feels very abrupt and I am not sure why these results are being presented here. The figure is nice, but is it necessary? Cant this be incorporated into a previous figure/section? |
| Table 3 | Is this the same information presented in Fig. 16? Eliminate if so. |

482-486    These conclusions were never clear to me in the main text. I think they get lost in a sea of other information presented.

---

## Author Response (AR1)

Title: Flow Structure and Mixing Near a Small River Plume Front: Winyah Bay, SC, USA

Author(s): Christopher Papageorgiou et al.

MS No.: egusphere-2025-189 MS type: Research article

**Authors Response to Reviewer Comments**

My co-authors and I would like to express our sincere gratitude to Dr Spicer (Reviewer 1) and the anonymous Reviewer 2 for their thoughtful and constructive review. We revised the manuscript following their recommendations. We believe we have responded adequately to all suggestions / requests. Although a thorough re-reading of the revised manuscript would be required, we also provide a summary of the changes we made in this document. by first listing each reviewer's comment and then our response using blue fonts.

A general suggestion by both reviewers was to focus more on what is new and highlight the new science / ideas that the manuscript has to offer. As both reviewers acknowledge the data set collected is of high quality. Our primary objective in the original submission was to present the data to the community without "biases" from our own interpretation. In retrospect this might have been a naïve approach, and we thank the reviewers for pointing this out and providing us with an opportunity to rectify. Major new topics we expand on the revised manuscript are: return flow under the plume, divergence of vertical flow at the interface of the counteracting flows, limited if any mixing behind the plume, potentially due to straining etc. We hope that this manuscript inspires our modeler colleagues to look at these processes in more detail.

We hope that this revised version is acceptable for publication in EGU Oceans.

Sincerely

George Voulgaris

**Specific Responses to Reviewer 1 (Preston Spicer, PNNL)**

Comments from Reviewer 1 (Preston Spicer, PNNL)

This is an excellent data set with great potential as a valuable contribution to the river plume community. My general comment is there is a lot going on in this manuscript: the figures are busy, and there are many redundancies. I find myself often losing track of what is important and what is extra "fluA". I think the results, discussion, and associated figures can be streamlined and the story

made clearer. I am generally fine with the analysis, but some relatively significant restructuring and rewriting is needed. I tried to specify my individual concerns below.

**Major Comments:**

1. Introduction: there are not clear knowledge gaps outlined, so it is not clear if this data set and study are novel. Other folks have studied turbulence and mixing around plume fronts...what is new here? Make it clearer.

An attempt has been made to make it clearer, although in the introduction we can identify the problem, but we do not reveal the results and what is new. We discuss the new methods used and expanded the introduction by adding references to other people's work.

2. The tidal dynamics section (3.5) needs more clarifying information.

More clarification has been provided in the text and a figure has been added with the spatial distribution of the stations used for the tidal analysis presented.

3. Restructuring associated with explaining results is needed. There are simply too many figures / too much text that are often redundant. It is very easy to get lost when reading from the amount of data.

Although the figures were reduced by 1 some of them were redacted and the order was revised (especially regarding the acoustic backscatter).

4. Similarly, there are lots of turbulence/mixing parameters presented but it is not clear if they are all contributing to the story. Mixing is probably the most important part to this paper, but I feel the authors' do not guide the reader properly through their train of thought. Some variables appear, are not really discussed, or simply don't matter. I think there is also a major lack of discussion on dynamics/processes with too much focus on describing patterns in turbulence. Look through the figures and text and only present what is needed for your story. Repackage it.

The reviewer is correct, mixing, or lack thereof, is part of the story. In the revised version the discussion of the development of return flows under the plume has been expanded. This has been made more comprehensive, especially after we elected to rotate the flows into an across- and along-front component. We do believe that all figures presented merit to be presented as they provide the essence of the analysis used (Smith, 2020). It is true that some plots seem to be busy, however there are clear patterns in there that we call the reader to focus and not on the details of the individual data points. We hope that the message gets across clearly and we do not believe that we need to force the reader into "our" conclusions through super averaging a smooting.

5. The elevated currents and turbulence at depth after plume front passage is very interesting, but I don't think the dynamics associated with this are ever addressed. Why do the currents speed up so much at depth after the front passes? Is bottom boundary mixing (Spicer et al.,

2021 JPO; Whitney et al., 2024 JGR) important? This has not been observed in many systems and would be very impactful/novel to quantify here.

**References:**

Spicer et al., 2021: The EAect of Bottom-Generated Tidal Mixing on Tidally Pulsed River Plumes, Journal of Physical Oceanography, <a href="https://doi.org/10.1175/JPO-D-20-0228.1">https://doi.org/10.1175/JPO-D-20-0228.1</a>

Whitney et al., 2024: Mixing of the Connecticut River Plume During Ambient Flood Tides: Spatial Heterogeneity and Contributions of Bottom-Generated and Interfacial Mixing, JGR: Oceans, <a href="https://doi.org/10.1029/2023JC020423">https://doi.org/10.1029/2023JC020423</a>

We thank the reviewer for the suggestions and references. Tidal signal is present but is limited to the bottom boundary layer. This is now made clearer in the text. There is not evidence of tidal mixing of the plume because of the tides, and this has been discussed in the revised manuscript.

**Minor Comments (by line number):**

53 Says "near surface mixing" here but bottom layer mixing is mentioned in the abstract.

It has been corrected to total water column.

Fig 1. Do you have bathymetry contours that could be added to Fig. 1a? This would be very helpful to the reader.... particularly if bottom boundary mixing is discussed in the paper.

Fig 1 has been updated as suggested by the reviewer.

99 How far away, exactly, was the vessel?

The vessel was more than 150 m away. It has been included in the text.

108-121 We did a very similar upriser-mode MicroCTD sampling scheme in the Merrimack River plume (Spicer et al., 2022):

https://agupubs.onlinelibrary.wiley.com/doi/full/10.1029/2022GL099633. It may be useful for you to cross-reference our paper for differences in turbulence estimation parameters. Also, did you take multiple profiles are average them together? Or just use singular profiles?

All data shown are single profiles. The averaging is one when individual profiles are averaged into their corresponding groups (A to E).

144-146 I do not understand this logic. Why such a difference in discharge estimates? How far away is the USGS gauge?

This has been explained in the text. There are more rivers discharging into the Winyah Bay estuary that are not instrumented; the 1.5 coefficient is to account for them, and it is based on previous studies.

146 Why not show the tides? Seems important for a plume front that forms with the tide.

Tidal elevations and river discharge have been included in Fig. 2

148 Is this low water in reference to low water at the NOAA gauge or at Winyah Bay

At the NOAA tide gage which is further away although the phase lag is less ~ 15-20 min. This is included in the text.

151 Can you use nearby NOAA wave buoys to back this up.

Not nearby wave bouys, we were using our own visual observations. We updated the values using NOAA NDBC station 41024 that is some 70 miles away but in the nearshore. This does not affect our findings.

Fig. 2 I think it would be good to show river discharge and tidal elevations as well so we can see how the sampling day conditions fall in to the broader period.

Tidal elevations and river discharge have been included in Fig. 2

You should mention the number of casts and when they were taken (i.e., a group before the front, a group in the front, a group after the front) in the methods section above.

Mentioned in the methods now too, however they are still shown in Fig 3b in Results.

Label the groups in Fig. 3b. Perhaps color the vertical lines marking the profiles the same colors as those in Fig. 4.

The groups have been colored as suggested.

Fig. 4, 5 Why is group A warmer and fresher than group B?

Most likely because of the position of the small boat at that time. Significant drifting was occurring during the microCTD deployments, that in reality this temporal variability includes some spatial variability too. Good observation, but we do not think this alters the data interpretation.

Fig. 6 Why not make contour plots with time on x-axis, depth on y-axis, and velocities colored? If the ADCP was in a fixed position, this makes the most sense. That would explicitly show time variation and be a little easier to interpret/align with Fig. 3 time variation.

Done as suggested. Furthermore, the currents were rotated into across-front (u') and along-front (v') velocity components as this makes interpretation easier. See text for details.

After reading through this section more, I feel strongly this should be a contour plot. Although the current profiles show an obvious increase in plume layer velocities, it is very hard to distinguish what velocities are positive and negative here particularly in the sub-plume layers. Contour plots with a red blue colormap would be ideal and helpful.

Red blue colormap has been used as suggested. It improved presentation significantly.

206-208? Mark the front arrival time on the velocity plot/contours.

Done as suggested.

A map of where these stations are located should be given in an appendix or supplemental material. What are these 80 stations? 80 ADCPs set up at the same time? This seems excessive and unlikely. More information is needed.

More information is supplied in the text and a map has been included in Fig. 8 as requested.

230 Why 4.5 m? It would appear the plume influences currents below that depth in Fig. 6.

We did experiment with different depths and there were not significant differences by going deeper, so this depth was selected.

231 Why only these two harmonics?

It is explained in the text now. We the data available (only 5 days) we can safely resolve only one diurnal and one semi-diurnal constituent (Raleigh criterion). We did not include subharmonics (e.g., M4) as these are of secondary importance offshore.

Fig 7. The velocity jumps evident at hour 0 in the inset would indicate there is plume influence here and the 4.5 m threshold is not the best to use. You refer to these jumps being a result of plume influence near bottom. Was the total depth at T2 ever mentioned? (yes it is 11.5m) If the average includes plume velocities near 4.5 m, then I wouldn't call this near bottom. What do these scatter points look like if the threshold is deepened to 5 or 6 m? (tried but did not make much of a difference) How deep is the water column at all the stations used here? Does it vary? By how much?

The water depths at the other stations varied from 7 to 14m. That variation justified the selection of 4.5 m as to ensure enough data for depth averaging for the tidal analysis.

If the current changes direction exactly when the plume front passes, I don't think you can call this the tidal current...going back to my comments above. This is some combination of tide and plume. At all the other stations used to predict the tidal current, are they within or outside the plume? Why do we not see these jumps elsewhere? What accounts for the outliers in Fig. 7a and b which deviate significantly from the predicted tidal curve?

All other stations were collected at different times and not at the time of the plume experiment. Some locations were within the geographical location of the plume but not at the same time as the data collected at TS2.

The differentiation between tidal current and plume-influenced current is clearer now that the data are rotated in the across- and along-front coordinate system. The deviation of the flow from the predicted tidal current is the influence of the plume. Given that the across front tidal component is the weakest, the majority of the deviation seen in Figure 6a is attributed to the plume.

Fig. 8 You probably don't need the TKE dissipation rate colorbar going as low as 10^-12.... this is essentially unimportant noise. I would constrain to 10^-10 or even 10^-8 to better emphasize where the strong turbulence is. Also, the MSPE panel is probably not necessary as long as you explain in the text why data is missing, which you have. Going back to my

comment on making contours of the current velocities, you should add them to this figure (after removing MSPE) and moving this to Fig. 6. Then you can describe currents and associated turbulence together. Alternatively, you can replace MSPE with contours of shear squared ( $S^2 = \frac{du}{dz^2} + \frac{dv}{dz^2}$ ) to check how vertical velocity shear aligns with turbulence.

MSPE has been removed and the TKE dissipation range has been adjusted as suggested. Also, the shear  $(S^2)$  is plotted in the same plot.

Lastly, make the colormap of the vertical velocities red-blue so we can easily identify what is upward and downward moving velocity.

**Done as suggested.**

263-270 If this is true, please add an appendix or supplemental material to confirm to the reader. Typically, MicroCTD estimates are more reliable than those from an ADCP. It is strange to throw out those profiles here.

The particular profiles are shown later in Figure 15. The reference to the thesis was for a more generic comparison of the two methods. We do not believe that any method is better than the other as turbulence measurements are challenging. The main difference is that the one provides averaged values over time, while the other provides an instantaneous snapshot of turbulence for each profile.

Further, the thesis being referred to sharing this information has a link that does not work.

The link has been double checked and it seems to work: https://scholarcommons.sc.edu/etd/7640/

Fig 10. Panel (a) should be at the top with succeeding panels beneath. I see now the shear squared which is good. I still think shear should be shown with turbulence, as they are linked. I don't think the current vectors are helpful on this map (as it is not oriented with the cardinal directions) and are better shown as red-blue colormaps as I described previously. Also, showing u and v here again is redundant to Fig. 6. Combine like-things rather than describe them twice in two different places.

Current vectors have been removed as suggested and shear plot has been moved together with the Structure Function TKE dissipation estimates – see Fig. 9

305-325 Haven't current speeds and direction already been discussed? This sections is quite redundant and should be integrated into the results where velocities were already discussed.

**Done as suggested.**

This delta U\_L1-L2 variable is not clear. Is it the difference in U\_L1 and U\_L2....it doesn't seem that way considering it remains well above 0 in Fig. 10.

This plot has been removed to avoid confusion. The new discussion on the vertical flow structure, using the new coordinate system we believe has simplified the narrative.

- Can you describe why the frontal Reynolds number is being calculated. What is the relevance to your story.
- It mainly for other readers to use it if they want to scale their results to ours. It can be removed if you insist but for now, we have retained it.
- Table 1 If this information is presented in Fig. 10, then it seems redundant to include in a table as well.

We removed the figure, so we retained the table.

357 Equation for buoyancy frequency?

It has been included in the revised manuscript in the introduction.

- Interpolation should be from the finer resolution to coarser: i.e., N^2 interpolated onto S^2 depth vectors.
- Using central differencing to calculate gradients in a coarse grid creates some problems, especially in intense vertical structure conditions. The issue has been resolved by not calculating Rig but following the Spicer et al (2022) approach and plotting 4N2 and S2 values together.
- Fig 12. You already showed S^2 in Fig. 10. These are more unnecessary redundancies which take up figure space.
- See above, also this is for the profiles and for getting a sense of the Rig. As such is required in this updated figure.
- 370-378 Ri can be greater than 0.25 and still have intense mixing. This is particularly true for convective instability driven mixing, where often Ri >1, and we would expect to be happening in the front (see Ivey et al., 2020: https://agupubs.onlinelibrary.wiley.com/doi/full/10.1029/2020GL089455). I think your Ri profiles need to be smoothed more to get helpful info. By interpolating onto the coarser ADCP depth vectors rather than the finer CTD depths, this should help.

Thanks for this recommendation, this has been included in the manuscript.

Eq. 3-5 Kind of an odd order to presenting these equations. Equation 3 uses Equation 5, so it makes more sense to have Equation 5 come before 3.

Order has been changed as suggested.

Fig 13. This is too much to look at. Can you average by group and not show every single profile? This would be helpful here and in other figures. Where are the depth labels on the bottom panel?

The objective here is not for the reader to follow each individual profile. The eye is the best smoothing algorithm and I think the data clearly cluster for the reader to objectively interpret without being guided by smoothing.

420-425 I am interested how your cleaned up Ri profiles compare to this metric. Ri can indicate where convective or shear mixing is expected as well. It would be valuable to present the two parameters together and discuss alignment or lack thereof.

The two analyses are presented now with the Rig estimates being converted to mixing efficiency  $\Gamma$  as in Spicer et al (2021) and are compared against the Smith (2020) method. They do not agree and this is discussed in the text.

420-430 Again....is 10 m "near-bed"? I don't think total depth was ever mentioned.

It has been mentioned in a couple of places in the text (~ 11.5 m) and also shown in the velocity profile transects figure.

Fig 14 ADCP TKE dissipation estimates were already presented way before this in Fig. 8....where it makes sense to show comparisons with the other TKE dissipation estimation methods. Combine or remove to eliminate more redundancy.

This comparison requires profiles; the previous plot shows the advantage of the AD2CP in providing high temporal resolution something not possible with the micro-structure sensors. Again, we could not see how to compare a contour plot with a profile, thus we have elected to leave as is.

Elaborate on why these mixing efficiency values are notable. This would imply convective mixing, correct?

This section has been enriched and elaborated, we argue more on turbulence reduction due to straining reducing mixing.

This whole backscatter section feels very abrupt and I am not sure why these results are being presented here. The figure is nice, but is it necessary? Cant this be incorporated into a previous figure/section?

It has been moved after the figure showing the currents.

Table 3. Is this the same information presented in Fig. 16? Eliminate if so.

We eliminated figure 16 and retained the table.

482-486 These conclusions were never clear to me in the main text. I think they get lost in a sea of other information.

This section has been re-written making a better connection with the findings presented in the main text.

**Specific Responses to Reviewer 2**

**Reviewer 2 Comments**

The manuscript presents field measurements of water-column mean and turbulent statistics during a passing river plume. The topic falls in the scope of the journal Ocean Science. The presentation could be improved to more effectively emphasize the novelty of the study and the science behind these unique measurements. I would recommend publication provided that my suggestions are considered.

**Major suggestions:**

Introduction: Suggest more explicitly highlighting the novelty of this study. Currently, the authors provided a review of relevant existing studies. Still, it is vague what the knowledge gaps are left in those studies and how this new set of measurements will address any unresolved questions.

Section 4.4: While the Figs. 12 to 16 are adequately described, how each variable is calculated, how its value changes, etc., the implied dynamics and physics are not discussed in detail. Suggest strengthening the discussion. Also, Figures 14 and 15 seem very noisy, and the scattering of the data is not addressed. I suggest the authors improve the clarity of the figures.

Conclusion: Suggest better highlighting the most important findings of the paper. Please also consider discussing possible future work.

As we indicated in response to Reviewer 1, the manuscript has undergone major revision and we have attempted to better highlight the most important findings as suggested. With regards to the figures, we have referenced Smith (2020) for the calculations (we actually used the code supplied by the author as supplementary documents) we provided only a brief description of the principle behind the calculations. The noise in the data is real and as we mentioned in our response to Reviewer 1 the objective in here is to look at the grouping and emerging patterns and not on individual points..

**Minor suggestions:**

Fig. 1(b): what is the variable shown in the image?

Not clear what "variable" the reviewer refers to as this is a satellite image of the study site showing the plume. We speculate the reviewer might be referring to TS2, which is the station location.

Fig. 3(c): The unit of dT/dz is missing.

This has been corrected now.

Fig. 6(c) and Line 220: the vertical current ~ cm/s is quite strong. It also indicates horizontal divergence above the depth of maximum w and convergence below. I would suggest the authors more clearly explain the circulation.

We have updated the figure to now show both the vertical velocity from the 4 beams (broad band) and the more accurate vertical velocity from the 5th beam of the ADCP. The 5th beam velocity is less sensitive to conversions of the radial, along-beam velocity to ENU coordinates. The signs remain the same although the magnitude is slightly reduced. Also the circulation is better explained in this version, after we elected to present the currents in a coordinate system aligned with the direction of front propagation. See updated figure and text for details.

Fig. 8(c): suggest using a different colormap to better highlight positive and negative, for example, lowbluehighred.

Done as suggested.

Line 262: the sentence is not clear to me. Suggest clarifying.

In the revised manuscript this sentence has been removed.

Line 273: suggest deleting "As presented earlier,"

Done

Fig. 9: what does the purple arrow in Layer 2 mean?

Indicates propagation direction of the front / new plume, it has been removed in the revised manuscript to avoid confusion.

Fig. 10: suggest changing the order of the panels so that panel a is at the top and panel d is at the bottom.

Done as suggested.

Table 3: in the caption, please define \epsilon\_k.

Corrected

Fig. 14: the unit for K\_\{rho} is wrong.

Corrected.

---

## Referee Report (RR1)

**Preston Spicer, PNNL**

**Major Comments:**

I think this paper is GREATLY improved since I first read it, but there are still more improvements needed.

- 1. I still struggle to follow the mixing aspect of this paper and what story is being presented there. There still does not seem to be a clear message and conflicting analyses are presented (i.e., the mixing efficiency) but never discussed or reconciled except for a few broad sentences at the very end of the conclusions (which is not a good place for it). Identifying mixing mechanisms is an objective of the paper, but I feel accomplishing this objective is muddled in the text. It is there...it just could be clearer.
- 2. Tidal straining is mentioned in the introduction and conclusions, but there does not seem to be any analysis on the mechanism in the rest of the paper. Either do not mention tidal straining or add a much more detailed analysis of the process.

**Minor Comments (by line number):**

- In the text "...a return was flow developed...", the "was" should be removed.
- It would be helpful to expand a bit on the concept of tidal straining here. Even just a few more sentences describing the basics of it.
- 142-143 Looks like a few lines repeat here.
- Fig. 1 Looks much better with bathymetry!
- 145-158 Were other MicroCTD quality control flags utilized besides the terminal velocity? How about instrument inclination? What was the range of speeds allowed around the terminal velocity? See Spicer et. al (2023) (Evolving Interior Mixing Regimes in a Tidal River Plume) Supporting Information for things I am looking for.

I am sorry to nit-pick. But since individual profiles are being used and not being averaged together, data post-processing is important for this study.

Fig. 2 The caption for panel (c) is written in kind of a confusing way. Tidal elevation is black line. Solid red is recorded discharged and dashed red is tidally corrected discharge...correct?

Fig. 6 Looks much better. To really make it easy, you could label direction directly on the figures or colorbar: i.e., blue on top panel = traveling with front, blue in second panel = towards coast, etc.

Also, directly label which panel corresponds to u' and which is v'. Having both variables on the colorbar makes it less clear. I understand it is in the caption, but make it clearer so readers do not necessarily need to read the caption to know.

You mean across-front here, right?

You are saying the Garvine model explains these vertical current patterns? Or that frontal convergence in general drives vertical currents? Clarify....these lines are a little opaque.

You are not showing divergence on these plots. If you are going to make claims about where different types of divergence are correlated, you should plot it. You could mark lines where dw/dz = 0 and du/dx = 0. The addition of divergence is good but it seems half-explained at the moment.

Fig. 8 Label the map panel (c) or (d). It is the only subplot without a label.

Looking at Fig. 9, there doesn't seem to really be a difference in shear at depth before and after front passage. It is a bit strange that dissipation is increasing.

You are not calculating a correlation coefficient, so you should not say there is a correlation.

359 "Extend" should be "extent".

Table 2 Caption...specify each variable and it's long name to reminder readers and provide a thorough description of the table.

Also...why are units cm/s here and m/s everywhere else?

Provide more information on why you are presenting these nondimensional numbers.

Also, for the Froude number varying with time in Fig. 11, is the layer depth varying with time as well? Or just velocity? You only mention an average Froude number in the text which is why I ask. Make sure to mention the range in values shown in your Fig. 11.

| Similar point for frontal Reynolds number. Also, why is 0.36 m/s used for Uf |
|------------------------------------------------------------------------------|
| for that number while it appears Fr is calculated with Uf = 0.61 m/s?        |

Is "U\_s" in Fig. 11 the same as "U\_f" in the text?

| 397     | Where does 0.33 m/s come from?                                                                                                                                                                                                                                                                                |
|---------|---------------------------------------------------------------------------------------------------------------------------------------------------------------------------------------------------------------------------------------------------------------------------------------------------------------|
| 401     | There is a stray "t" before "LW".                                                                                                                                                                                                                                                                             |
| 443     | You have 4N^2 plotted, not 4S^2.                                                                                                                                                                                                                                                                              |
| 446     | Label each subplot by group number in Fig. 13.                                                                                                                                                                                                                                                                |
| 446-457 | This paragraph could be streamlined to just focus on most important patterns. It is a little confusing to read right now.                                                                                                                                                                                     |
| 458     | It is not too surprising that shear exceeds stratification in the bottom layerthere is no plume there. This is pretty common.                                                                                                                                                                                 |
| 458-467 | After reading this paragraph, you could easily remove the paragraph directly before this.                                                                                                                                                                                                                     |
| 481-499 | This is goodmy only comment is you mention "Smith (2020)" in most sentences. Probably not necessary and would read more natural if not mentioning the reference so much.                                                                                                                                      |
| Eq. 6   | Please explain physically what the turbulence potential energy (TPE) is and why it is compared to TKE in Fig. 15.                                                                                                                                                                                             |
| Fig. 14 | The y-axis tick labels are missing from the bottom plots.                                                                                                                                                                                                                                                     |
| 525-530 | Mixing efficiencies calculated by the TPE to TKE ratio (Table 3) obviously differ from those determined by the flux Richardson number (Fig. 13). In fact, they present nearly opposite messages about where mixing is important. I do not see any text here reconciling that. This brings up a few questions: |
|         | Why is mixing efficiency calculated in multiple ways? (this is never said).                                                                                                                                                                                                                                   |
|         | Which is correct?                                                                                                                                                                                                                                                                                             |
|         | Does comparison between the methods provide insight into mechanisms?                                                                                                                                                                                                                                          |
| 533     | Typo in this line. "we have not estimates of" needs to be reworded.                                                                                                                                                                                                                                           |
| 549     | Typo: "e_k was of the order of"?                                                                                                                                                                                                                                                                              |
| 555     | Not according to Fig. 13.                                                                                                                                                                                                                                                                                     |
|         |                                                                                                                                                                                                                                                                                                               |

- There are a handful of grammatical errors in this paragraph which should be addressed. I also struggle to follow what is important here. The authors bounce between nondimensional numbers but do not make clear why. What is notable about the decrease in diffusivity after frontal passage? Is this surprising?
- I thought the lower layer differed significantly from what we would expect the tide to do (i.e., Fig, 8)?
- Again...Fig. 13 shows high mixing efficiencies throughout the top layers for all groups. So you have conflicting results which need to be addressed.
- 580 Spicer et al. (2021) was an idealized modeling study so no observations in the CT river plume.
- This is the first mention of the discrepancies in results. This should be in the discussion and expanded on more....not saved for the last paragraph of the conclusions.

Further, the idea of straining as a mixing/stratifying mechanism is mentioned in the introduction then again here in the conclusions.... but I don't think there is any analysis or discussion on straining in the remainder of the text. If there is, I seem to be forgetting it. Either completely remove the idea of straining from this paper or provide a true analysis of the mechanism.

---

## Author Response (AR2)

Title: Flow Structure and Mixing Near a Small River Plume Front: Winyah Bay, SC, USA

Author(s): Papageorgiou et al. MS No.: egusphere-2025-189 MS type: Research article

Once more we appreciate the time reviewer Spicer spent on this manuscript and the constructive comments provided. These have been considered and fully incorporated in the revised manuscript.

Responses to each comment are presented below (in blue fonts) while the file of the manuscript with the tracking on provides more details on where the changes have been carried out.

We hope that this revised version is acceptable for publication in EGU Oceans.

Sincerely

George Voulgaris

**Authors Response to Reviewer Comments**

**Major comments:**

1. I still struggle to follow the mixing aspect of this paper and what story is being presented there. There still does not seem to be a clear message and conflicting analyses are presented (i.e., the mixing efficiency) but never discussed or reconciled except for a few broad sentences at the very end of the conclusions (which is not a good place for it). Identifying mixing mechanisms is an objective of the paper, but I feel accomplishing this objective is muddled in the text. It is there...it just could be clearer.

We have clarified the objective and findings of the mixing efficiency story. We agree with the reviewer that the mixing data are conflicting. In the revised manuscript the conflict is reconciled through the concept of the layered turbulence in highly stratified flows as presented in the review paper of Caulfield (2021). Under highly stratified flows, like in our data, turbulence can be high in certain regions with weak stratification while other regions of the flow are characterized by low intensity turbulence but higher

stratification. However, the limitation of resolving the spatial distribution of TKE dissipation rates due to the averaging methods used does not allow this layering to be identified while the high dissipation rates of the high turbulence regions dominate the recorded total signal. The concept described above explains the reduced mixing found within the plume and the higher dissipation rates. We present the results of a standard parameterization used in modeling to demonstrate the differences and this estimation has been moved after the explanation / discussion of the observed data and the purpose of these estimates is clearly stated now.

2. Tidal straining is mentioned in the introduction and conclusions, but there does not seem to be any analysis on the mechanism in the rest of the paper. Either do not mention tidal straining or add a much more detailed analysis of the process.

In the revised paper it has been clarified that tidal straining presents macro-scale conditions like what we encounter, but as per its definition the terms describes the driving force but not the mixing mechanism per se. It is argued that the mechanism is the same as explained above (layered turbulence) and straining is the driver that creates the highly stratified conditions. We have also added some text explaining the concept of tidal straining.

**Minor comments (by line number):**

In the text "...a return was flow developed...", the "was" should be removed.

Done

It would be helpful to expand a bit on the concept of tidal straining here. Even just a few more sentences describing the basics of it.

Done as suggested

Looks like a few lines repeat here.

Corrected

Fig. 1 Looks much better with bathymetry

We agree thanks for the suggestion

Were other MicroCTD quality control flags utilized besides the terminal velocity? How about instrument inclination? What was the range of speeds allowed around the terminal velocity? See Spicer et. al (2023) (Evolving Interior Mixing Regimes in a Tidal River Plume) Supporting Information for things I am looking for.

I am sorry to nit-pick. But since individual profiles are being used and not being averaged together, data post-processing is important for this study.

The concerns expressed are reasonable, and we appreciate the desire to be clear about how the data were quality controlled. Additional information on the screening process has been provided, only one data point had to be removed due to high angle of attack or high vibration (likely due to interference with the line from the weight), and no pair of dissipation estimates from the shear probes differed by more than one order of magnitude, so no points were removed due to those criteria. The minimum speed allowed was 0.5 m/s, and maximum vertical velocities were 0.7 m/s. Given the strong density changes, some variability is to be expected.

The general agreement between the microCTD and ADCP dissipation estimates is further confirmation that our values are reasonable. As an additional check, a downward-profiling VMP 250 was deployed from the vessel simultaneously with the microCTD. These values are affected by the ship-generated turbulence in the surface layer, but show good agreement at depth (results presented in Papageorgiou 2023). I'm including the figure here showing the comparison for the relevant stations, much of the near-surface VMP data had to be removed due to high angle of attack and/or vibration, and there is likely still contamination due to the ship, causing some of the bias down to 6 m. Red is VMP and black is microCTD.

Fig. 2 The caption for panel (c) is written in kind of a confusing way. Tidal elevation is black line. Solid red is recorded discharged and dashed red is tidally corrected discharge...correct?

It is fixed by adding colors to caption description

Fig. 6 Looks much better. To really make it easy, you could label direction directly on the figures or colorbar: i.e., blue on top panel = traveling with front, blue in second panel = towards coast, etc.

Also, directly label which panel corresponds to u' and which is v'. Having both variables on the colorbar makes it less clear. I understand it is in the caption, but make it clearer so readers do not necessarily need to read the caption to know.

The figure has been modified as suggested.

You mean across-front here, right?

Yes, it has been clarified in the text

You are saying the Garvine model explains these vertical current patterns? Or that frontal convergence in general drives vertical currents? Clarify....these lines are a little opaque.

The Garvine model refers to the shallowing of the base of the front and not on the flow convergence / divergence. This is argued to be a secondary effect because of that shallowing. If the base of the plume was horizontal (no shallowing) then du/dx at the z of the base will be zero. It should be noted that in the revised paper the convergence has been revised; it is divergence. This has been updated in the text and discussion of the vertical flows is discussed accordingly.

You are not showing divergence on these plots. If you are going to make claims about where different types of divergence are correlated, you should plot it. You could mark lines where dw/dz = 0 and du/dx = 0. The addition of divergence is good but it seems half-explained at the moment.

We have no information on du/dx this is inferred from the plot of du/dt at the z where u'(z) = 0. We have made lines where u'(z) = 0 and the elevation of this line appears to be deeper than the w divergence line. The latter is easily identified by the white contour in the plot where w = 0.

Fig. 8 Label the map panel (c) or (d). It is the only subplot without a label.

Done as suggested

Looking at Fig. 9, there doesn't seem to really be a difference in shear at depth before and after front passage. It is a bit strange that dissipation is increasing.

While the shear on the scale of these figures is relatively low, stratification is also quite low, and so it may not take much shear to de-stabilize the lower part of the water column, where tidal currents (lower amplitude) are dominating the velocity signal. Furthermore, the concept of the turbulence layer invoked later on might explain this discrepancy,

You are not calculating a correlation coefficient, so you should not say there is a correlation.

Although correlation does not mean correlation coefficient, for clarity "correlation" was replaced with "relationship".

"Extend" should be "extent".

Fixed, thank you!

Table 2 Caption...specify each variable and it's long name to reminder readers and provide a thorough description of the table.

Done

Also...why are units cm/s here and m/s everywhere else?

This is due to the small magnitudes of the numbers we could replace if the editorial staff thinks that this is worth it, otherwise we would prefer to keep it as is.

Provide more information on why you are presenting these nondimensional numbers.

In the text we note that this is to provide information on the dynamics of the plume not only at the front but also behind it.

Also, for the Froude number varying with time in Fig. 11, is the layer depth varying with time as well? Or just velocity? You only mention an average Froude number in the text which is why I ask.

Yes the, estimation considers the layer depth too, as per the equation presented in the text. That's why the two lines do not mirror each other.

Make sure to mention the range in values shown in your Fig. 11.

A mention is made in the text that the Fr values are consistently above 1. The exact values are easily read on the plot itself.

Similar point for frontal Reynolds number. Also, why is 0.36 m/s used for Uf for that number while it appears Fr is calculated with Uf = 0.61 m/s?

Using D=2.2m g'=0.034 m/s2 and Uf=0.36 m/s we get  $1.316 \sim 1.32$ . If Uf=0.60m had been used then the value would have been 2.23. Uf=0.36 m/s is the relative velocity of the entire plume layer, rather than just the near-surface speed, which is faster (as shown in Fig. 11).

Is "U s" in Fig. 11 the same as "U f" in the text?

No U s is the absolute surface current / speed This is defined in the text now as well

Where does 0.33 m/s come from?

This is an averaged plume velocity over 2 hours after the front arrival. This has been clarified in the text now.

There is a stray "t" before "LW".

|     |     |   | - 1    |
|-----|-----|---|--------|
| ы   | 137 | 0 | $\sim$ |
| 111 | I X |   | u      |

443 You have 4N^2 plotted, not 4S^2.

Thank you for catching this, the figure is correct but the text had an incorrect formula, with N and S flipped in the gradient Richardson number. This has now been corrected in the text.

Label each subplot by group number in Fig. 13.

We do not think this is necessary. The subplots of the lower raw have the groups that correspond to the subplot on the top row

This paragraph could be streamlined to just focus on most important patterns. It is a little confusing to read right now.

The paragraph has been re-written to focus on the most important points regarding trends in buoyancy and shear.

It is not too surprising that shear exceeds stratification in the bottom layer....there is no plume there. This is pretty common.

This has now been acknowledged in the text.

After reading this paragraph, you could easily remove the paragraph directly before this.

The paragraph has been re-written and we elected to leave it in the paper.

This is good...my only comment is you mention "Smith (2020)" in most sentences. Probably not necessary and would read more natural if not mentioning the reference so much.

This has been addressed as suggested by the reviewer.

Eq. 6 Please explain physically what the turbulence potential energy (TPE) is and why it is compared to TKE in Fig. 15.

A definition / what is the TPE has been added following earlier on, after eq 2 where TPE appears for the first time. As shown in there this is the "theoretical" definition of the mixing efficiency (i.e., kinetic energy required to destroy the potential energy the stratified flow processes). A sentence reminding the reader that the ratio of TKE dissipation rate over TPE dissipation rate is used to defining mixing efficiency has been added in the paragraph describing Fig 15 too.

Fig. 14 The y-axis tick labels are missing from the bottom plots.

Not sure more ticks are needed, in addition the grid-lines clearly show the different tick levels.

Mixing efficiencies calculated by the TPE to TKE ratio (Table 3) obviously differ from those determined by the flux Richardson number (Fig. 13). In fact, they

present nearly opposite messages about where mixing is important. I do not see any text here reconciling that. This brings up a few questions:

Why is mixing efficiency calculated in multiple ways? (this is never said).

This has been addressed now (see our response in major comments). The one is parametrization the other is based on overturning scales. We believe the latter one is more appropriate and we have explained our rationale in the revised paper.

Which is correct?

**See above**

Does comparison between the methods provide insight into mechanisms?

As explained in the revised text the reason for the parameterized values is to demonstrate the contradictory results someone would obtain using this method.

Typo in this line. "we have not estimates of" needs to be reworded.

Fixed

Typo: "e k was of the order of"?

Fixed

571

Not according to Fig. 13.

This part of the manuscript has been re-written as explained in our response to the major comments above.

There are a handful of grammatical errors in this paragraph which should be addressed. I also struggle to follow what is important here. The authors bounce between nondimensional numbers but do not make clear why. What is notable about the decrease in diffusivity after frontal passage? Is this surprising?

In here we describe the observations.

I thought the lower layer differed significantly from what we would expect the tide to do (i.e., Fig, 8)?

Since the tidal velocities are increasing over the data collection period, it makes sense that shear is also increasing and therefore bottom-induced mixing is elevated as the tidal velocities strengthen.

Again...Fig. 13 shows high mixing efficiencies throughout the top layers for all groups. So you have conflicting results which need to be addressed.

We believe this has been addressed see response to major comments

580 Spicer et al. (2021) was an idealized modeling study so no observations in the

CT river plume.

Language adjusted to reflect this.

This is the first mention of the discrepancies in results. This should be in the discussion and expanded on more....not saved for the last paragraph of the conclusions.

Further, the idea of straining as a mixing/stratifying mechanism is mentioned in the introduction then again here in the conclusions.... but I don't think there is any analysis or discussion on straining in the remainder of the text. If there is, I seem to be forgetting it. Either completely remove the idea of straining from this paper or provide a true analysis of the mechanism.

We have expanded on these in the revised manuscript as suggested. See our response in major comments and revised paper.

---

## Author Response (AR3)

Title: Flow Structure and Mixing Near a Small River Plume Front: Winyah Bay, SC, USA

Author(s): Papageorgiou et al. MS No.: egusphere-2025-189 MS type: Research article

Dear Editor,

Once more we appreciate the time reviewers spent on this manuscript and the constructive comments provided during the process.

The final version of the accepted manuscript is submitted here. The manuscript have been thoroughly checked for editorial and grammatical errors and my co-authors and I believe that is ready for publication in EGU Oceans.

Thank you for the opportunity

Sincerely

George Voulgaris

On behalf of all my co-authors